# Potential and challenges for sustainable progress in human longevity

Florian Bonnet [1], Ina Alliger[2], Carlo-Giovanni Camarda[1], Sebastian Klüsener [2,3,4], France Meslé[1], Michael Mühlichen [2], Josselin Thuilliez [5] & Pavel Grigoriev [2] ✉

Decelerating gains in life expectancy ($e_0$) in high-income countries have raised concerns about the future of human longevity. To enhance our understanding of these developments, we examine subnational ($N = 450$) mortality trends in Western Europe in the period 1992-2019. Between 1992 and 2005, gains in life expectancy were both substantial and widespread. Laggard regions experienced the fastest improvements, yielding rapid regional convergence. Between 2005 and 2019, however, gains in these regions decelerated, while remaining remarkably stable in vanguard regions, suggesting that it remains possible to continue extending longevity. The observed slowing of $e_0$ gains is strongly associated with mortality at ages 55-74, which increased in this period across large areas of Western Europe, particularly in Germany and France. In this work, we show that monitoring mortality trends at a fine geographical level is crucial for revealing both the potential for, and challenges to, sustainable progress in human longevity.

Life expectancy is a key indicator of public health and societal well-being, providing insights into the overall quality of life. While countries with high levels of income per capita (thereafter high-income countries) have generally enjoyed rising life expectancy over recent decades, in recent years, gains have been modest: stagnating mortality trends were reported[1-3] even before the adverse effects of the COVID-19 pandemic on mortality trends[4-6]. These developments have raised concerns about the sustainability of further progress in human longevity and the prospects for extending life expectancy in the future.

Potential limits on the extension of life expectancy have been a long-standing subject of intense scientific debate[7]. Two decades ago, Oeppen and Vaupel[8] argued that these limits were far from being reached, since 'female life expectancy in the record-holding country has risen for 160 years at a steady pace of 3 months per year'. Examining trends in life expectancy from 1990 to 2019 for the eight countries where it was highest in 2019, Olshansky and colleagues[9] concluded that 'radical human life extension is implausible in [the 21st] century', as life expectancy gains have considerably declined over the last three decades. Using mortality data from 16 countries with high life

expectancy, Dowd and colleagues[1] pointed out the uncertainties surrounding the future of life expectancy, in a context of slowing gains. A more recent cohort analysis of 23 high-income countries supports these conclusions, finding again that life expectancy gains are decelerating[10]. In this general context, some individual countries have attracted growing attention in recent years. This is particularly true for the United States, where rising drug-related mortality and the failure to reduce cardiovascular mortality have been identified as the major drivers of the stagnation in life expectancy in the 2010s[2]. High cardiovascular mortality in the US has also been identified as the main reason for its unfavourable position compared to other high-income countries[11,12]. The same applies to England and Wales, where life expectancy has diverged from the rest of Europe, stagnating since the mid-2010s[13]. Another prominent case is Germany, where life expectancy gains have recently slowed[14].

Inevitably, analyses of mortality conducted at the national level have largely overlooked significant regional variations, which present a more detailed and nuanced picture[15]. One of the notable exceptions is a study using county-level data by race for the US to define 'ten

[1]French Institute for Demographic Studies (Ined), Aubervilliers, France. [2]Federal Institute for Population Research (BiB), Wiesbaden, Germany. [3]Institute of Sociology and Social Psychology (ISS), University of Cologne, Cologne, Germany. [4]Vytautas Magnus University, Kaunas, Lithuania. [5]French National Centre for Scientific Research (CNRS), Rennes, France. ✉e-mail: pavel.grigoriev@bib.bund.de

Americas' with different life expectancy trends since 2000[16]. It showed that the life expectancy of whites in low-income Appalachia and the lower Mississippi Valley has stagnated since 2000, while the life expectancy of Asian Americans continued to rise even during the 2010s. Rashid and colleagues[17], analysing data from 6791 communities in England, showed that female life expectancy fell between 2014 and 2019 in 1,270 communities, and male life expectancy in 784 communities. Mühlichen et al.[18] reported substantial regional variations in Germany, with some regions making substantial progress since the 1990s and others struggling to keep pace. While they provide valuable insights, these studies only look at one or a limited number of countries.

In this work, we substantially extend existing knowledge through a comprehensive analysis of life expectancy trends in 450 subnational regions covering 13 Western European countries (representing a total population of approximately 400 million people in 2019) over the period 1992-2019. This period presents an ideal time window, covering a relatively stable time before the global COVID-19 pandemic, which heavily impacted life expectancy in European regions[19,20]. Western Europe, with its great diversity of demographic, economic, and institutional contexts, is very well suited for such an analysis. Notably, it includes several of the countries with the highest levels of life expectancy, such as Spain, Italy and Switzerland. Another advantage of Western Europe is that detailed and reliable long-term mortality data stratified by year, region, sex, and age are generally available. Nevertheless, the data routinely collected by national statistical offices are far from homogeneous and cannot be directly used for comparative analyses (Supplementary Table 3). We thus begin by harmonizing the data to ensure a uniform structure and the comparability of spatial units over the observation period. We then apply a state-of-the-art demographic methodology[21,22] to smooth the data and calculate mortality indicators. We use life expectancy at birth, a widely used and understood, easily interpretable mortality measure, as the main mortality outcome. It also has the advantage of providing more stable statistical estimates compared to partial life expectancy or life expectancy at specific ages, which is a particularly valuable feature for small spatial units. Finally, its use facilitates comparisons with other national and regional studies.

The main added value of our analysis lies in its extensive geographical coverage and high spatial resolution. By analysing smoothed, long-term mortality trajectories across a contiguous set of regions in Western Europe, we can unravel emerging trends and patterns that are often obscured in national-level analyses. We particularly focus on disparities in pathways towards longevity limits observed between vanguard and laggard regions. These complementary concepts of vanguard and laggard populations have often been used in demographic research to describe diverse mortality trajectories at the subnational level. There has been a worldwide general trend towards widening health inequalities by socioeconomic status (SES), which is measured by different dimensions, including the level of educational attainment. In general, the population groups with higher SES enjoy faster declines in mortality and gains in life expectancy than the groups with the lowest SES[23]. Substantial and increasing inequalities in mortality by marital status have also been reported[24]. Our analysis contributes to debates on the possibility of increasing human longevity, as vanguard populations pave the way towards higher life expectancy for other groups[7]. Finally, identifying geographical longevity hot and cold spots can deepen our understanding of trends in life expectancy and inform targeted public health interventions, including better allocation of healthcare resources.

## Results
### Evolution of longevity gains
Figure 1 offers a dual perspective on annual gains in life expectancy at birth (hereafter life expectancy or $e_0$) across 450 regions in Western Europe from 1992 (the earliest year for which data are available for all of the countries included) to 2019 (the last year before the COVID-19 pandemic). The x-axis represents the average level of these gains, while the y-axis represents their variability, quantified by their standard deviation. This visualization provides a general overview of the evolution of annual life expectancy improvements and how equitably they have been distributed across European regions.

Two distinct periods can be identified. From 1992 to the mid-2000s, average annual gains in both male and female life expectancy remained remarkably stable, as indicated by the nearly vertical trend lines. During this period, gains in male life expectancy were larger (gold circles), averaging approximately 100 days per year (more than three months), while average gains in female life expectancy (purple triangles) were significantly smaller, at about 72 days per year. At the same time, regional variability in life expectancy gains was highest in the early 1990s, with standard deviations of 35 days for males and 28 days for females. This variability declined over the subsequent years, reaching a nadir in 2004-2005. Over this time, the pace of longevity improvements had become more homogeneous across European regions, with standard deviations decreasing by nearly 50% for both sexes and converging to approximately 15 days.

After this decade of improvements, the mid-2000s marked an important turning point away from longevity trends in Western Europe. Since then, life expectancy gains have steadily declined, while regional heterogeneity has gradually increased, especially for males. By the early 2010s, annual gains had dropped to 80 days for males and 50 days for females. These declines continued during the 2010s, falling to just 55 and 35 days respectively in 2018-2019. This represents approximately a two-fold reduction in life expectancy gains compared to the 1990s.

In Fig. 2, we build upon the averages and standard deviations presented in Fig. 1 by displaying the life expectancy trends for vanguard and laggard regions. We define vanguard and laggard regions as the top and bottom deciles of the distribution, respectively. Additionally, we show trends in the maximum and minimum life expectancy observed annually across all spatial units. The solid black line between the vanguard (blue) and laggard (red) lines represents the average life expectancy for all regions combined. Table 1 gives the average life expectancy gains for the entire period as well as for each decade, specifically for these groups of regions. Supplementary Table 1 provides the same indicators but for the periods 1992–2004 and 2005–2019.

The solid black line, which flattens through the 2010s for both males and females, illustrates the general deceleration in longevity improvement during the last decade of the study period, as illustrated in Fig. 1. However, the patterns for the vanguard and laggard regions differ notably. The trends for the laggard group, as well as those for the region with the lowest female life expectancy in each year (the lowest line in the right-hand side panel of Fig. 2), show a more pronounced decline in life expectancy gains compared to the overall average. The values in Table 1 provide additional quantitative evidence for this visual representation. For laggard regions, male and female life expectancy gains decreased from 129 days (roughly four months) and 94 days (three months) per year, respectively in the 1990s to 59 days (two months) and 39 days (roughly one month) per year in the 2010s. For the minimum life expectancy region, male and female gains decreased from 129 days (roughly four months) and 94 days (three months), respectively in the 2000s to 105 days (three months and a half) and 45 days (one month and a half) in the 2010s.

By contrast, the upper two lines in Fig. 2 in both panels, representing the vanguard group, show no clear indication of a decline in life expectancy gains. This distinction from laggard groups is even more pronounced for the maximum life expectancy region, where the data reveal nearly linear upward trajectories, indicating relatively stable longevity gains over these three decades. The figures in Table 1 reflect a slight reduction in both male and female life expectancy gains

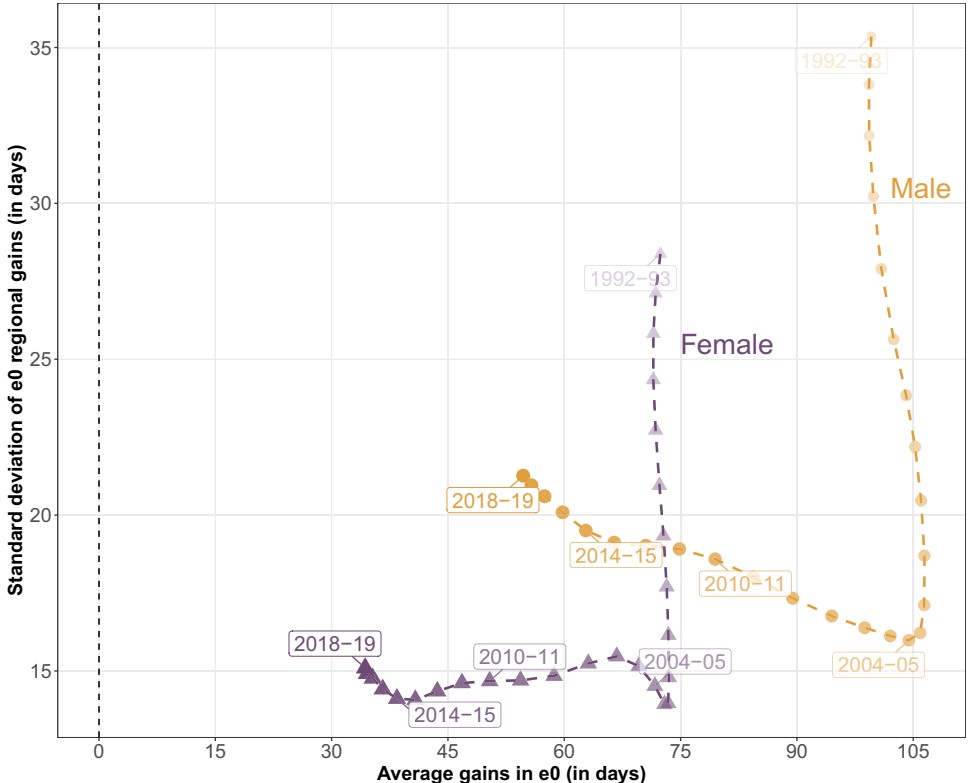

**Fig. 1 | Pathways of mean annual regional gains in male and female life expectancy ($e_0$) and their variance across 450 Western European regions, 1992–2019.** Average gains in $e_0$ and the standard deviation of regional $e_0$ gains are unweighted. Each point corresponds to values obtained from $e_0$ regional gains between the first and the second year. Source data are provided as a Source Data file.

in the vanguard regions, from 3 and 2 months per year, respectively in the 1990s to 2.5 and 1.5 months per year in the 2010s. Meanwhile, gains in the record-holding regions remained relatively constant, at approximately 2.5 (male) and 1.5 (female) months annually.

Across the three decades of the study period, the maximum and minimum levels of male and female life expectancy occurred in nine different European regions. Four Spanish regions (Araba/Alava, Guadalajara, Salamanca and Soria) largely dominated male or female longevity in Western Europe. The two exceptions to this Spanish dominance are the Swiss region of Zug, which held the record for male life expectancy between 2007 and 2018, and the western part of inner London, which held it in 2019. The regions with the lowest life expectancies in a given year include two Portuguese regions, the Azores (both male and female in different periods) and Madeira (male), as well as Scotland (female).

We continue by exploring the relationship between life expectancy ranking and gains in more detail. Each dot in Fig. 3 represents a specific region, positioned according to its rank in the life expectancy distribution (x-axis) and its life expectancy gain in a given period (y-axis). The three selected periods, 1992–1993, 2004–2005 and 2018–2019, are distinguished by colour. The solid lines depict smooth functions for each period. The first- and last-time intervals correspond to the beginning and end points of our study period, and 2004–2005 represents the pivotal point in the life expectancy gains pathway seen in Fig. 1. Note that as all regions are ranked according to the estimated value of life expectancy at the beginning of a period, the ordering of individual regions (their location on the x-axis) varies between the different periods.

In 1992–1993, it was the laggard regions that saw the largest gains in both male and female life expectancy, as indicated by the cloud of green points in the upper right corner of Fig. 3 (both panels). During the same period, the vanguard regions (the top decile for life

expectancy) experienced substantially lower gains. In contrast, the orange line for the second period (2004–2005) is relatively flat across the ranking for both sexes. This suggests that there was no clear relationship between ranking and life expectancy gains in 2004–2005. Finally, in 2018–2019, the relationship reversed for males, as indicated by the downward-sloping violet line in the left panel of Fig. 3. This indicates that, in contrast to the earlier periods, vanguard regions experienced higher gains in male life expectancy than the laggard regions in the late 2010s. The same is true for female life expectancy, although the difference between vanguard and laggard regions is smaller. The shift to a positive relationship between life expectancy and longevity gains is largely due to a significant (approximately fourfold) decrease in life expectancy gains at the lower end of the longevity distribution between 1992 and 2019. Annual gains in life expectancy at the upper end of the distribution, in contrast, remained relatively stable across the three periods.

## Geography of longevity gains

Figure 2 reveals the regions with the highest and lowest life expectancies in different years within the study period. Our next analyses explore geographical disparities in Western Europe in greater detail. Figures 4 and 5 map regional male and female life expectancy gains (panels a, c and e) and ranking (panels b, d and f), in 1992–1993, 2004–2005 and 2018–2019. The vanguard (dark blue) and laggard (dark red) groups, shown in the panels b, d and f and denoted as D1 and D10, refer to the top and bottom deciles, and correspond to the groups of regions represented in Figs. 2 and 3. We also highlight the second (D2) and ninth deciles (D9) of the life expectancy distribution, in order to better understand the geographic clustering of laggard and vanguard regions. In Supplementary Fig. 1, we also show the average annual gains in life expectancy for the periods 1992–2004 and 2005–2019.

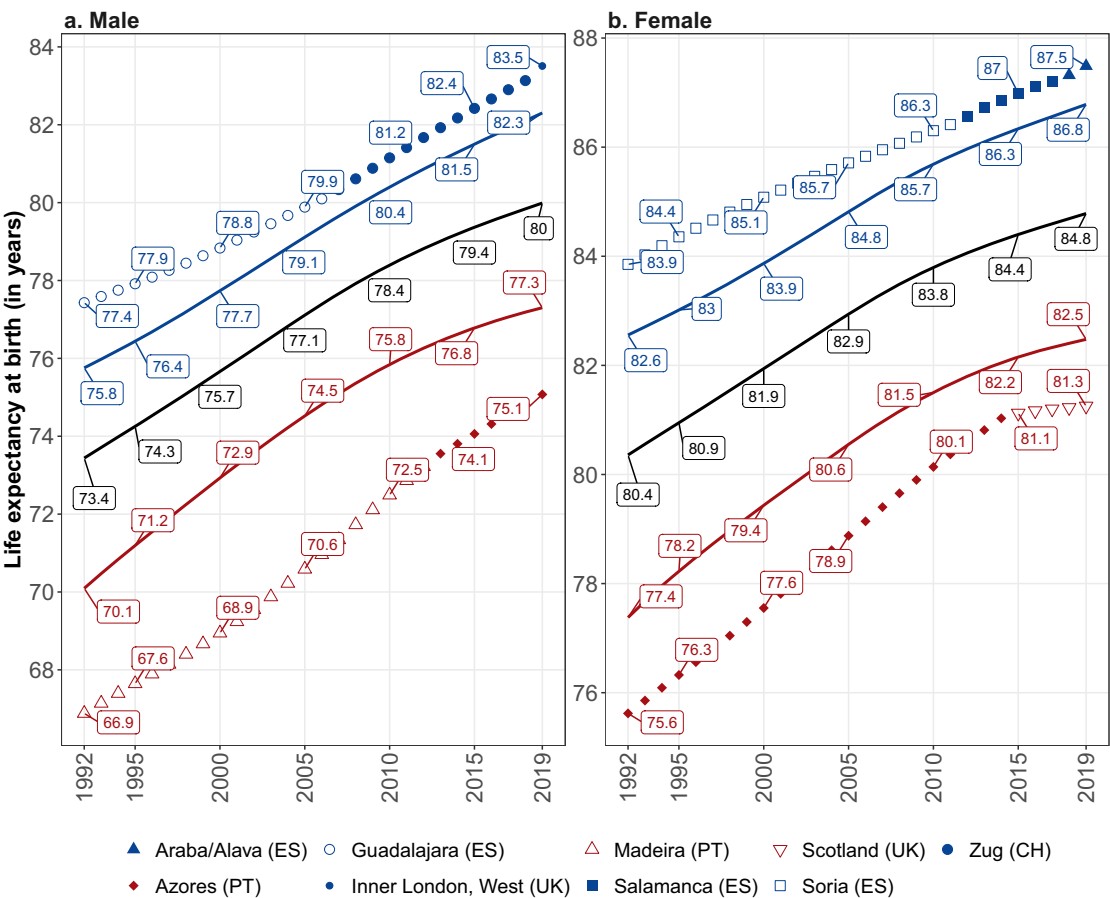

**Fig. 2 | Trends in life expectancy ($e_0$) in vanguard and laggard Western European regions, 1992–2019.** Trends in $e_0$ for Vanguard and Laggard Western European regions are shown for males (panel **a**) and females (panel **b**). The red (respectively blue) line shows the unweighted average $e_0$ of regions in the top (respectively bottom) decile of the regional distribution, and the black line shows the unweighted average across all 450 regions. Minimum and maximum values are indicated by distinct point symbols, corresponding to the respective regions. Source data are provided as a Source Data file.

**Table 1 | Annual life expectancy gains (in days) in vanguard and laggard Western European regions, 1992–2019**

|  | Male |  |  |  | Female |  |  |  |
|---|---|---|---|---|---|---|---|---|
|  | **1992–2019** | 1992–99 | 2000–09 | 2010–19 | **1992–2019** | 1992–99 | 2000–09 | 2010–19 |
| Maximum | **82** | 64 | 85 | 96 | **49** | 56 | 45 | 48 |
| Vanguard | **89** | 90 | 97 | 77 | **57** | 60 | 66 | 44 |
| Mean | **89** | 101 | 100 | 65 | **60** | 72 | 68 | 40 |
| Laggard | **97** | 129 | 106 | 59 | **69** | 94 | 75 | 39 |
| Minimum | **111** | 94 | 129 | 105 | **76** | 88 | 94 | 45 |

Values for vanguard and laggard regions are based on the unweighted life expectancy at birth of regions in each group. Values for the mean are computed from the unweighted life expectancy at birth across the 450 European regions in the panel. Values in bold are for the whole period. Source data are provided as a Source Data file.

Figure 4a, c and e shows substantial gains in male life expectancy in 1992–1993 and 2004–2005 across all regions in Western Europe. The largest gains in 1992–1993 were observed in eastern Germany and several northern Italian regions. By 2004–2005, the largest increases were still seen in northern Italy, as well as in several regions in eastern Germany, the UK and Portugal. By 2018–2019, however, life expectancy gains had slowed everywhere, particularly in Germany and the UK, with the exception of Northern Ireland and southern England.

The overall geographical ranking of male life expectancy appears relatively stable at first glance (see Fig. 4b, d and f). But important shifts occurred within national populations, particularly in regions that either led or lagged in life expectancy. In 1992-1993, laggard regions were mainly located in eastern Germany, Portugal, near the Belgian-French border, and Scotland. By 2018-2019, these clusters had expanded into western Germany and southern Denmark, persisted in the Belgian-French border region, and disappeared from northern Portugal. Apart from the cluster around the Belgian-French border, life expectancy gains in laggard regions became much smaller over time. Vanguard regions in 1992, on the other hand, were primarily found in Italy (excluding the north), central Spain, and southern England. Over time, this cluster expanded into northern Italy and Switzerland but retreated from southern Italy, parts of Spain, and southern France.

A similar pattern is observed for female longevity. Laggard zones expanded into western Germany, emerged in Wallonia (southern Belgium), and disappeared from Portugal, Sicily, and Denmark (see Fig. 5b, d and f). Life expectancy gains in today's laggard regions are

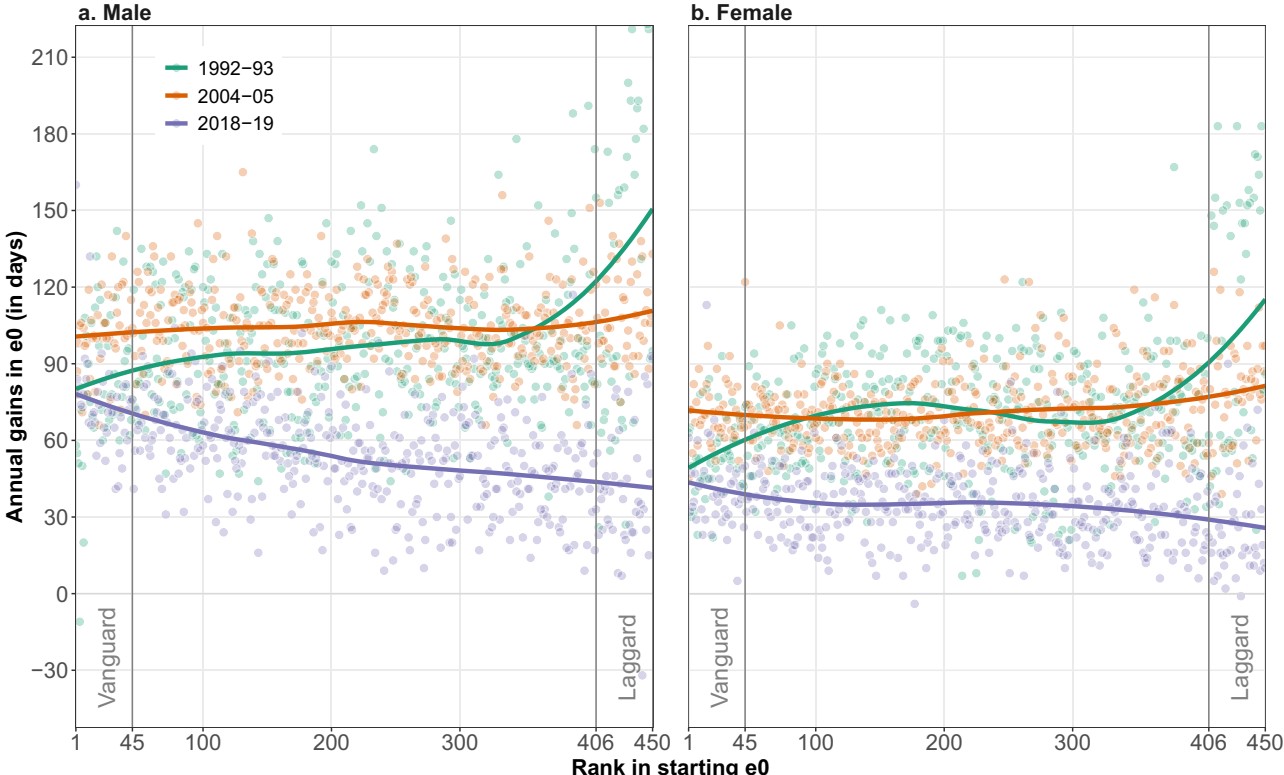

**Fig. 3 | Annual gains in life expectancy ($e_0$) by $e_0$ ranking across 450 Western European regions, in 1992–1993, 2004–2005 and 2018–2019.** Annual gains in $e_0$ by regional $e_0$ ranking are shown across the two panels. **a** depicts the relationship for males, where each point represents a region and corresponds to the change in $e_0$ between the first and second year of each period (1992–1993, 2004-2005 and 2018–2019). **b** depicts the same relationship for females. Smoothed lines were computed using local regression techniques. Source data are provided as a Source Data file.

significantly lower than those recorded in the former laggard regions (see Fig. 5a, c and e). The geography of vanguard regions for female life expectancy also changed over the study period. Over time, the number of regions among those with the highest female life expectancies in France dwindled, and by 2019, these vanguard regions were primarily concentrated in eastern Spain and northern Italy. For females as for males, absolute life expectancy gains in the vanguard regions of 2018–2019 were comparable to those observed in 1992–1993 for the original vanguard regions. Specifically, western France, northern Italy, and north-eastern Spain reported life expectancy increases of two months per year.

### Age-specific trends

Examining age-specific trends provides further insight into the observed patterns. Table 2 summarizes the average annual changes in the probability of dying in specific age ranges (35–54, 55–74, and 75–84) across all regions in our panel. The results are stratified by sex and decade, which enables us to identify the specific age groups and periods where mortality improvements were slowest. Supplementary Table 2 provides the same indicators but for the periods 1992–2004 and 2005–2019.

The results show that the pace of mortality improvements has particularly slowed between the ages of 55 and 74. While in the 1990s, female mortality in this age range was decreasing at 2.1% per year, in the 2010s, the rate was just 0.7%. And among males, the pace of mortality reduction at these ages declined from 2 to 1.2% per year over the same period. Notably, no such decline is observed for the other age groups. The pace of mortality reduction was steady at around 2% per year for ages 35 to 54 (slightly lower for female than male mortality), and about 1.5% per year for ages 75 to 84 (slightly lower for male than female mortality).

The spatial patterns of annual change in the probability of dying between the ages of 55 and 74 (20q55) can be seen in Fig. 6, for males (panels a, c and e) and females (panels b, d and f). Overall, a marked deceleration between 1992 and 2019 is evident. The decreasing pace of overall life expectancy gains is strongly associated to this deceleration (see Supplementary Fig. 2). Even more alarmingly, in several regions 20q55 actually increased at the end of the period (in light pink). This adverse trend is particularly pronounced among males in eastern Germany and females in western Germany. Conversely, the probability of dying between ages 55 and 74 continued to decline steadily in most of the vanguard regions, such as in northern Italy, Switzerland and eastern France.

The results of the analyses conducted for 20q35 and 10q75 appear in the Appendix (Supplementary Figs. 3 and 4). No negative mortality trends were observed in the 35-54 and 75–84 age groups. However, one important observation stands out: in the United Kingdom, the probability of dying between the ages of 35 and 54 increased during the final pre-pandemic years. This unfavourable trend is particularly pronounced for males, with the exclusion of southern England.

## Discussion

This study makes several potential contributions to the ongoing debate on life expectancy trends in high-income countries. Our study examines these trends using data at the level of subnational regions: in total, we cover 450 regions in 13 Western European countries. We believe that addressing life expectancy at a fine geographical level is paramount in understanding the potential to further improve human longevity, as national aggregates mask large differences within countries. For example, in France, there are stark contrasts between laggard regions in the north and vanguard regions in the south and east. The disparities between eastern and western Germany, and northern and

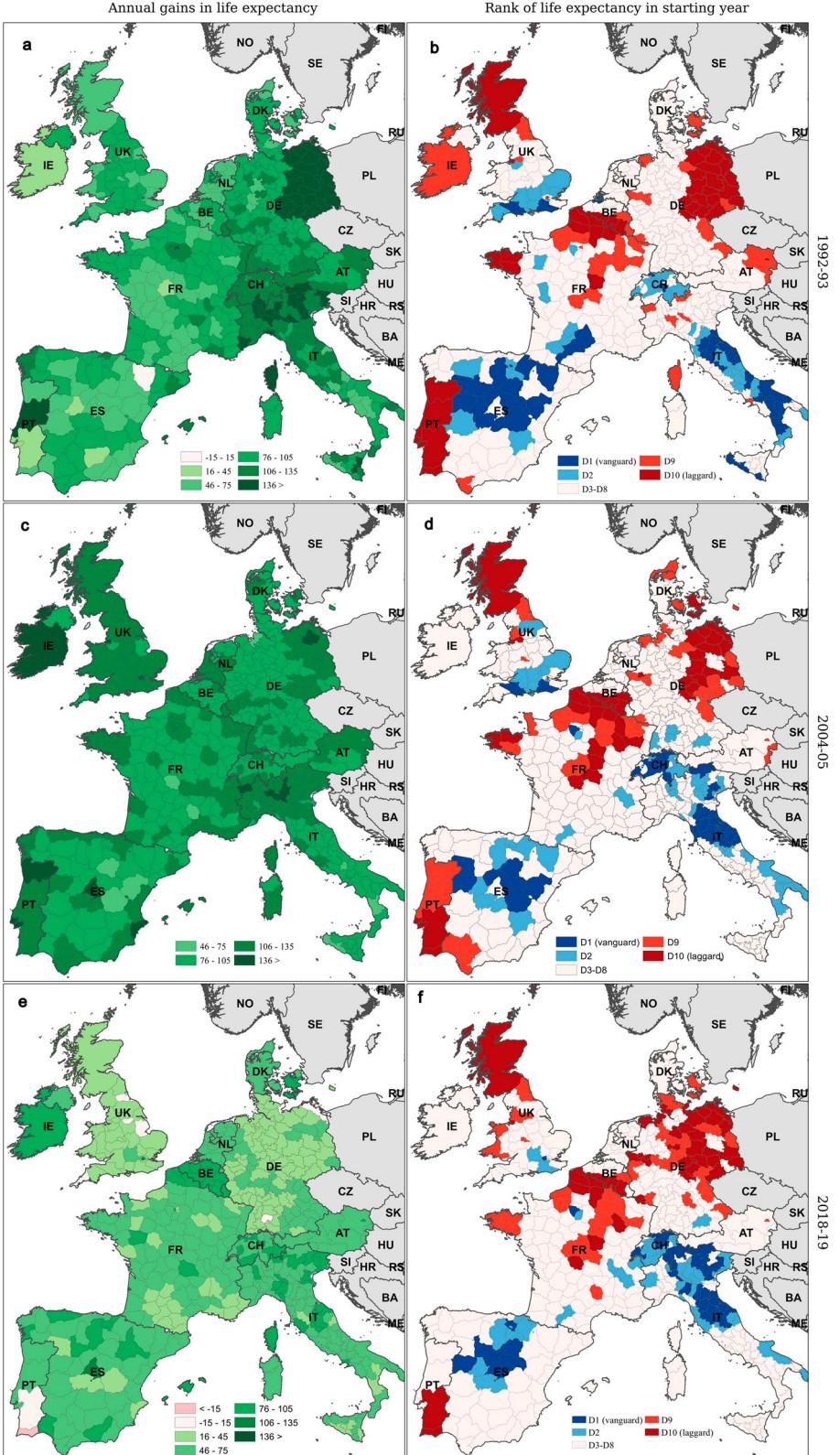

**Fig. 4 | Annual gains in male life expectancy ($e_0$), and ranking of 450 Western European regions for male $e_0$, in 1992–1993, 2004–2005 and 2018–2019.** Annual gains in male $e_0$ between the first and second year of the period and regional ranking are shown across the six panels. **a**, **c** and **e** provide the values of annual gains in $e_0$, panels **b**, **d** and **f** provide the ranks. For visual simplicity, the island regions of *Las Palmas* and *Tenerife* (ES) and the *Azores* and *Madeira* (PT) are not shown in maps in the paper. Source data are provided as a Source Data file.

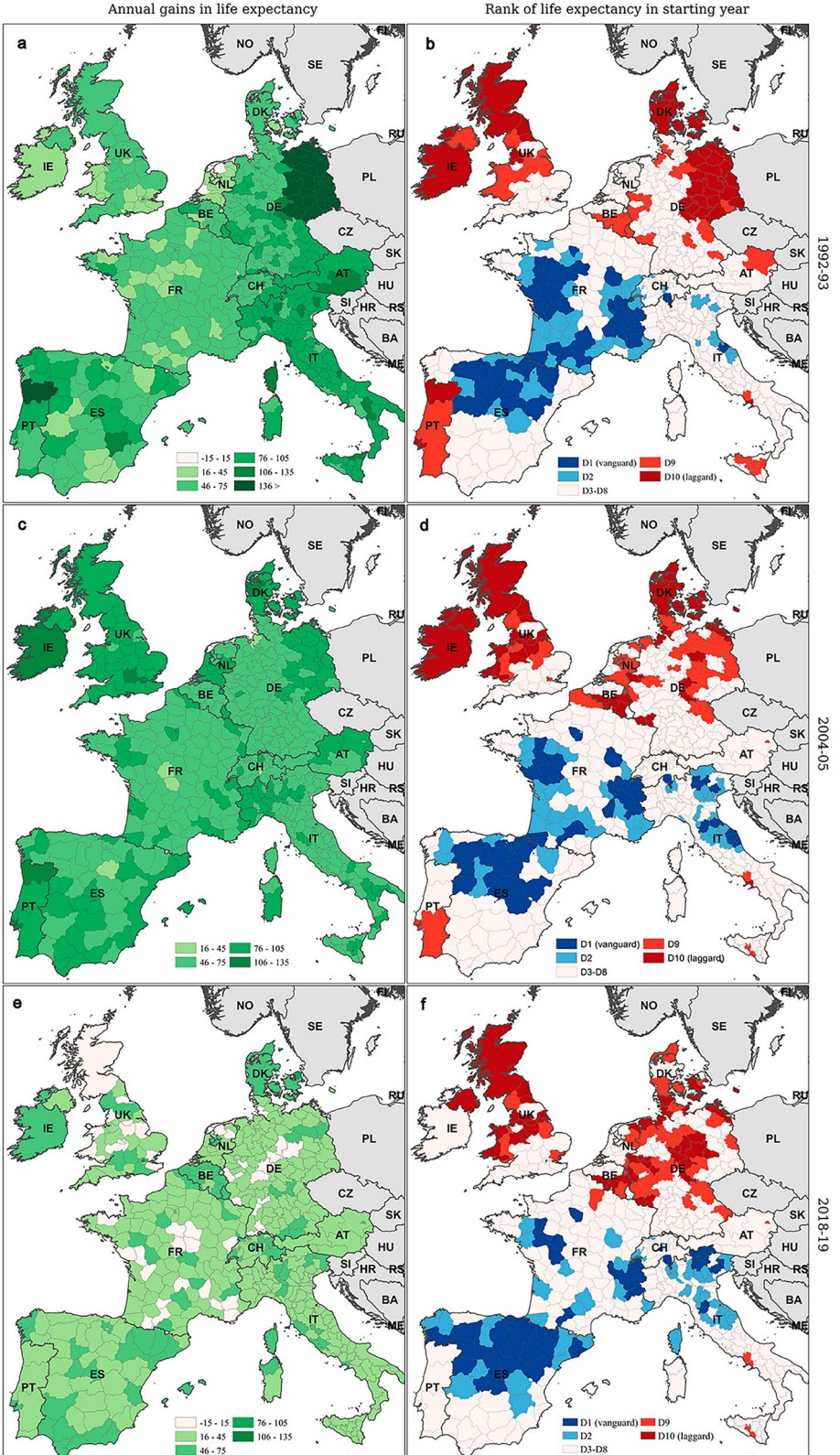

**Fig. 5 | Annual gains in female life expectancy ($e_0$), and ranking of 450 Western European regions for female $e_0$, in 1992–1993, 2004–2005 and 2018–2019.** Annual gains in female $e_0$ between the first and second year of the period and regional ranking are shown across the six panels. **a**, **c** and **e** provide the values of annual gains in $e_0$, **b**, **d** and **f** provide the ranks. For visual simplicity, the island regions of *Las Palmas* and *Tenerife* (ES) and the *Azores* and *Madeira* (PT) are not shown in maps in the paper. Source data are provided as a Source Data file.

**Table 2 | Average annual change (percent) in the probability of dying by age range across 450 Western European regions, 1992–2019**

|  | Male | | | | Female | | | |
|---|---|---|---|---|---|---|---|---|
| Prob. of dying between | **1992–2019** | 1992–99 | 2000–09 | 2010–19 | **1992–2019** | 1992–99 | 2000–09 | 2010–19 |
| 35 and 54 (20q35) | **−2.3** | −2.1 | −2.3 | −2.4 | **−1.7** | −1.6 | −1.7 | −1.8 |
| 55 and 74 (20q55) | **−1.8** | −2.0 | −2.2 | −1.2 | **−1.5** | −2.1 | −1.9 | -0.7 |
| 75 and 84 (10q75) | **−1.4** | −1.1 | −1.6 | −1.4 | **−1.6** | −1.5 | −1.8 | −1.4 |

The probability of dying between ages $a_1$ and $a_2$ is computed as the number of deaths between these ages divided by the number of individuals alive at age $a_1$ in the life table. Values in bold are for the whole period. Source data are provided as a Source Data file.

southern Belgium are equally pronounced. Together, they tell a compelling story of uneven regional progress. In addition to the empirical findings presented in this paper, we provide a data visualization tool for our longevity estimates at: https://histdemo.shinyapps.io/ReLoG_Europe/.

Our study identified two distinct phases in the evolution of life expectancy gains over the past three decades. The first phase, from 1992 to 2005, was characterized by stable and substantial life expectancy gains in Western Europe (about 2.5 months per year for females and 3.5 months per year for males). Over this period, the pace of gains across regions quickly converged. In contrast, the second phase, from 2005 to 2019, marked a period of declining life expectancy gains. By 2018–2019, annual gains had decreased to about one month per year for females and two months for males. This phase was also marked by a reversal of the previous convergence, with increasing regional divergence in the pace of gains across European regions.

Through an assessment of the pace of life expectancy gains in combination with a ranking of regions by life expectancy, then, our analysis reveals that the period 1992–2005 combined substantial overall gains in life expectancy with inter-regional convergence. During this 'golden era', it was laggard regions that made the greatest gains in life expectancy (4 months per year for males, and 3 months per year for females during the 1990s). These remarkable improvements significantly reduced the life expectancy gap between the worst- and best-performing regions. This convergence highlights the untapped potential for mortality reduction in laggard regions. Improvements in socioeconomic conditions, healthcare access and utilization, and individual health behaviour, together led to the rapid spread of the cardiovascular revolution, especially in eastern Germany after the fall of the Berlin Wall[25]. By contrast, the period 2005–2019 was much less favourable, as laggard regions saw shrinking gains in life expectancy: 40% lower than those observed during the previous decade and a half. Importantly, these gains were also slower, rather than faster, than those in the leading regions, which remained stable at around 2.5 months a year for males and 1.5 months for females. Our results show that these contrasting trends between leading and laggard regions resulted in a progressive mortality divergence across European regions, in line with previous research[26].

The driving forces behind this impressive reversal of fortunes can be better understood through the convergence-divergence framework, which explains the mechanisms leading mortality levels across populations to either converge or diverge. According to this theory, major innovations (e.g., drugs that reduce blood pressure) may initially trigger divergence, as some countries or groups are better positioned to benefit from them. Once access broadens, convergence tends to follow[27,28]. Importantly, convergence and divergence cycles do not necessarily alternate in a linear way; a new divergence process may unfold alongside an ongoing convergence process, complicating the interpretation of mortality trends. In our case, the convergence observed before 2005 is fully consistent with this framework. By contrast, the subsequent divergence was driven less by accelerated progress in vanguard regions than by a slowdown in laggard regions, a

pattern also observed in another recent study[29]. This observation does not call into question the validity or relevance of the convergence-divergence framework, but it highlights the difficulties facing laggard regions in maintaining momentum.

Our study was also able to highlight the evolving geography of the vanguard and laggard regions. For male life expectancy, the laggards were initially concentrated in eastern Germany, Portugal, regions near the Belgian-French border (Wallonia), and Scotland. Over time, they expanded into western Germany and southern Denmark, while northern Portugal saw relative improvement. On the other hand, vanguard regions were predominantly found in Italy (excluding the north), central Spain and southern England. Over the study period, they progressively spread to northern Italy and Switzerland, while retreating from southern Italy and parts of Spain. For female life expectancy, laggard regions expanded into western Germany and Wallonia, while continental Portugal, Sicily and Denmark improved their relative position. The number of vanguard regions in France decreased, and their concentration in eastern Spain and northern Italy grew.

This changing geography is crucial for understanding the potential for a longer life in Europe. Tracking life expectancy gains among the regions that constituted the vanguard in the 1990s reveals a slowing pace, and suggests possible limits to expanding human longevity. But this perspective overlooks the dynamic reranking of regions over time. When this reranking is taken into account, we observe that the pace of longevity gains has remained remarkably stable, particularly for the top-performing regions. This reinforces the thesis that human life expectancy has not yet reached its limits[8]. However, sustainable progress in longevity will require tremendous efforts and significant medical breakthroughs, particularly in reducing old-age mortality from aging-related diseases[4,30]. But the research agenda holds unprecedented promise, with great prospects for new scientific discoveries related to these aging-related pathologies on the horizon[31].

Looking beyond this optimistic view from the leading regions, however, the situation appears more worrying, as gains in life expectancy have declined sharply across Europe. This slowdown is strongly linked to the diminishing rate of decline in the probability of dying between the ages of 55 and 74. Moreover, the probability of dying has started to increase in large portions of Europe, particularly in eastern Germany for males and western Germany for females. These adverse trends can be partly explained by the long-term consequences of the transformation crisis of the early 1990s, which disproportionately affected young males[32]. The increase in female mortality in Western Germany is likely linked to the later onset of the smoking epidemic compared to other countries[33]. In contrast to these negative trends, age-specific mortality rates continue to decline in most of the vanguard regions, especially in northern Italy and Switzerland. This divergence highlights the need for targeted public health interventions to address emerging threats and mitigate these region-specific challenges.

A concerning development relates to young adults aged 35–54. The increase in mortality among this group appears to be driven by so-

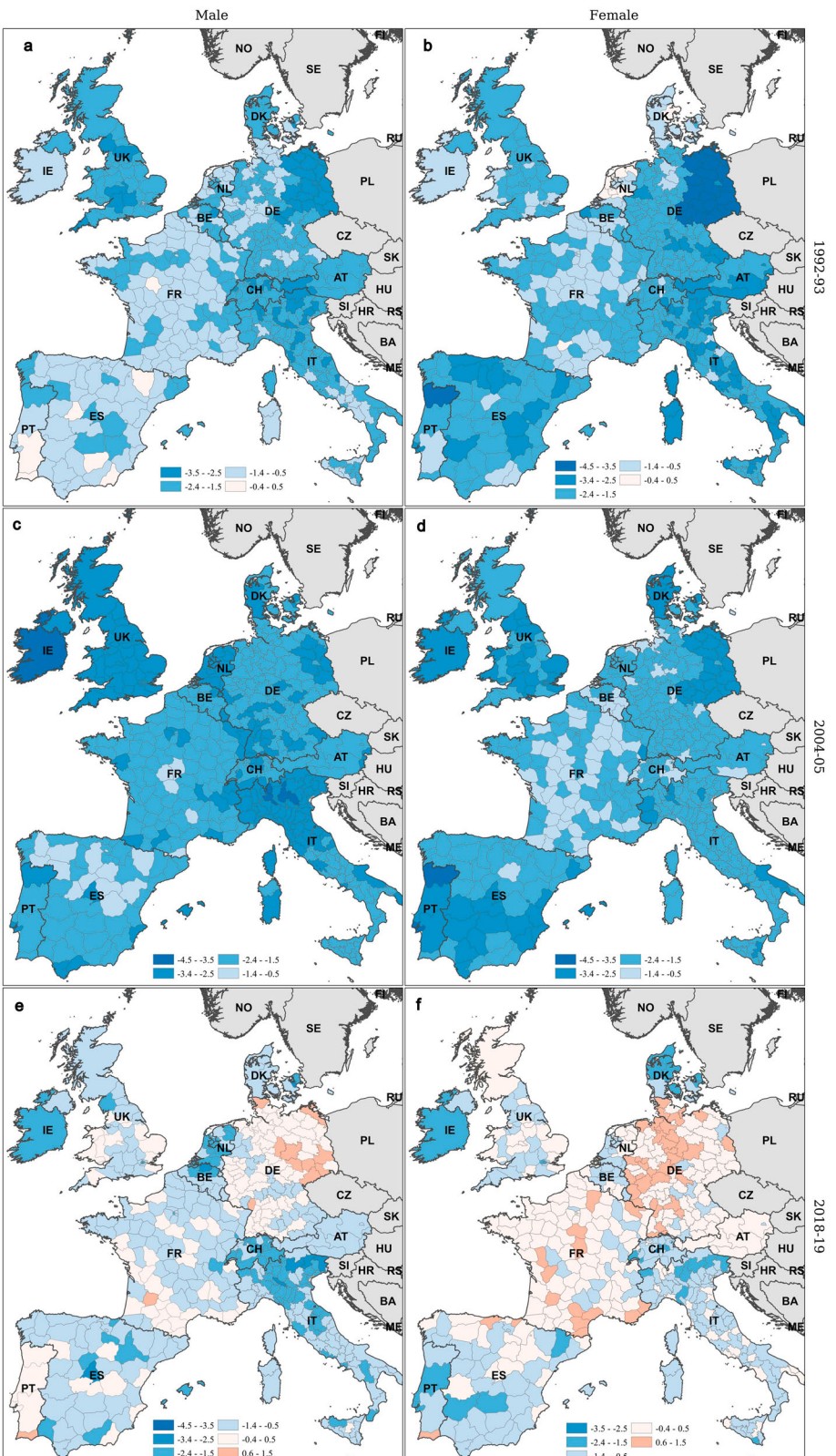

**Fig. 6 | Annual percentage change in the probability of dying between the ages of 55 and 74 for males and females across 450 Western European regions, in 1992–1993, 2004–2005, and 2018–2019.** Annual percentage changes in the probability of dying between ages 55 and 74 are displayed across the six panels for males and females. **a**, **c** and **e** show the annual percentage change for males between 1992 and 1993, between 2004 and 2005 and between 2018 and 2019, respectively, while **b**, **d** and **f** show the corresponding values for females across the 450 Western European regions. See Table 2 for more information about the variable. Source data are provided as a Source Data file.

called deaths of despair, from socially patterned causes such as alcohol, drug misuse and suicide[34]. Within the set of countries analysed here, this pattern has so far been documented only in the United Kingdom, with a particular concentration in northern regions. But it is plausible that similar clusters of elevated young adult mortality could emerge elsewhere in Europe in the near future. Such adverse trends at younger ages are already evident in other English-speaking countries. Indeed, it has been argued that the slowdown in life expectancy gains before the pandemic observed across these countries - with the exception of Ireland - largely stemmed from stagnating or rising mortality at young and middle adult ages[3].

Although our analyses are mainly aimed at providing empirical insights, and a detailed analysis of mortality determinants is beyond the scope of this study, it is important to highlight factors and potential causal mechanisms behind the observed subnational trends and differentials. Broadly speaking, it makes sense to situate the discussion of these factors within long-standing debates on the role of broad socioeconomic change versus targeted interventions in shaping mortality decline[35,36]. The classical McKeown thesis attributes rapid mortality decline in industrialized countries to broad economic and social changes, rather than to targeted public health or medical interventions. While this controversial thesis has since been overturned on empirical grounds, it posed the fundamental question of whether population health is determined primarily by broad-based social, political and economic conditions or by more targeted policies and interventions[37].

At the subnational level, this broader debate finds a close parallel in discussions of the relative role of individual versus contextual determinants of health[38] Accordingly, regional disparities in mortality can be interpreted in terms of compositional effects (reflecting the unequal distribution of individual risk factors across areas) or contextual effects (the influence of place-based conditions and policies). From a compositional perspective, our finding that stagnation in life expectancy has been most pronounced at ages 55–74 highlights the contribution of lifestyle-related factors such as alcohol consumption, drug use, poor diet and physical inactivity[1,39]. At the same time, contextual factors provide important complementary explanations. Recent research has emphasized the impact of austerity measures and so-called deaths of despair (suicide, drug misuse and alcohol-specific causes) on stagnating mortality in the UK[40–42]. In our study, the regional divergence in life expectancy observed since the mid-2000s broadly coincides with the period of the 2008 financial crisis and its aftermath, when economic growth slowed across Western Europe while becoming increasingly concentrated in major metropolitan regions[43,44]. These urban areas, which host high-skill service industries, are now among the best-performing regions in terms of longevity. For instance, between 2005 and 2019, the western part of central London rose from 132nd to 1st place for males and from 247th to 15th for females, while Paris advanced from 40th to 32nd for males and from 45th to 4th for females.

Looking ahead, emerging public health threats could slow or even reverse longevity improvements, even in regions with favourable socioeconomic conditions. Previous research has suggested that countervailing mechanisms, such as increased exposure to environmental risks and the diffusion of unhealthy behaviours may disproportionately affect higher-status groups[45]. Climate-related risks, notably heatwaves, are already having a measurable impact on mortality in European cities, with particularly strong effects observed in eastern Europe[46]. These factors are likely to become increasingly influential in the future, although well-designed policy interventions could mitigate their adverse effects on population health.

An important feature of our work is the combination of wide geographical coverage with high spatial resolution mortality series. We invested substantial effort to harmonize mortality data across regions and over time, ensuring comparability despite differences in national statistical practices (see Supplementary Table 3 for details on the implemented adjustments). This harmonization was essential to meeting our study's central objective: assessing long-term longevity trends both by individual region and by contrasting vanguard and laggard groups. Nonetheless, perfect comparability could not be achieved. The use of the NUTS classification, though standard in European demographic research, entails considerable variation in the population size of spatial units within the same NUTS level between countries. Moreover, while we aimed for the most detailed geographical units and the longest possible time series, differences in data availability obliged us to use less detailed information in some countries. This averaging of outcomes across larger areas may conceal inequalities in longevity, which recent studies have shown to be important for interpreting spatial disparities[47,48].

The extensive temporal coverage is another notable aspect of this study. Spanning nearly three decades prior to the COVID-19 pandemic, this study offers a unique opportunity to capture the evolution of life expectancy under relatively stable conditions. By leveraging and modelling long-term mortality data, we can disentangle short-term fluctuations from structural trends, and thus provide a solid foundation for understanding the slowdown in longevity gains. Our analysis deliberately focuses on the pre-pandemic period, consistent with earlier work on long-term mortality dynamics[1,9]. Including the pandemic years would have inevitably interfered with our assessment of structural trends, which was the main focus of this study. Consequently, our study does not take into account the mortality burden of COVID-19, which disproportionately affected vanguard regions in 2020 (e.g., central Spain, northern Italy, and the Paris region) and laggard regions in 2021[20]. Future research should extend the scope to incorporate both the pandemic and post-pandemic years, in order to assess whether and how this longevity crisis has reshaped trajectories of, and regional disparities in, life expectancy.

The design of our statistical methods is one of the defining features of our work. Because mortality estimates are sensitive to differences in population size between regions, smaller and more sparsely populated areas tend to display volatile and irregular mortality patterns, while indicators for larger regions are more stable. Such variability can hinder the establishment of clear conclusions. To address this issue, we used a P-spline model with smoothing parameters selected using the Bayesian information criterion (BIC), allowing the degree of smoothness to adapt to the characteristics of each region. This approach reduced the statistical noise typically associated with highly granular mortality data while preserving the underlying long-term patterns that are central to our analysis.

Further progress will require moving beyond aggregate life expectancy measures to investigate regional trends and patterns in cause-specific mortality. Such analyses would provide a more nuanced understanding of the drivers of regional divergence and convergence. In addition, identifying common characteristics (e.g., socioeconomic or cultural factors) that differentiate vanguard from laggard regions could shed more light on the reasons why some areas perform better than others. Addressing this question would eventually require systematic analyses that integrate mortality outcomes with area-level contextual variables. While this lies beyond the scope of our study, it represents a promising and necessary direction for future research.

With the observed deceleration of life expectancy gains, and particularly the prevailing mortality trends in Western Europe, concerns about the future development of human longevity are valid. Nevertheless, there are still reasons for optimism. First, the observed slowdown is not uniform across all regions. Vanguard regions in places such as central/northern Spain, northern Italy and Switzerland continue to push the boundaries of human longevity. Second, a similar situation unfolded in the 1960s, when many believed that the potential for further reductions in mortality had been exhausted after the near eradication of communicable diseases[7,8]. And yet the cardiovascular

revolution defied these expectations, leading to substantial further improvements. It is plausible that, once again, a new 'game changer' - such as advances in the prevention and treatment of cancer and dementia - could emerge to counterbalance the current unfavourable trends and drive future mortality reductions. Finally, we should not forget that over the 20th century, several attempts using sound argumentation were made to define the ultimate upper limit of human longevity[49–51]. Yet human lifespans have repeatedly surpassed these limits, pushing beyond what in the past was considered 'implausible' and 'overly optimistic'.

## Methods

### Data preparation

We collected subnational death and population counts for 13 Western European countries by age group and sex from Eurostat, the Human Mortality Database[52], and national statistical offices. To ensure the comparability of the selected spatial units in terms of size and structure, we relied mainly on the Nomenclature of Territorial Units for Statistics (NUTS), using NUTS 3 level units for Denmark, France, Italy, Luxembourg, Spain, and Switzerland, NUTS 2 units for Austria, Belgium, the Netherlands, Portugal, and England and Wales, and NUTS 0 units for Ireland, Northern Ireland and Scotland. For Germany, we used a national spatial classification (*Raumordnungsregionen*)[18]. Due to territorial changes over time and data availability issues, minor adjustments of the raw data had to be made (see Supplementary Table 3 for details).

In the cases of Italy and Austria, mortality data did not differentiate between deaths occurring under one year of age and those occurring between ages 1 and 4: only aggregated death counts for ages 0-4 were available. To refine our life table estimates, we disaggregated these deaths, assuming that the ratio of the mortality rate for children under one year old to that of children aged 1-4 was constant across all regions of a country and equal to the national-level ratio obtained from the Human Mortality Database[52]. This procedure was applied separately for each year and sex.

### Methodology

In this study, we model mortality over age and time for each spatial unit independently to ensure that region-specific mortality developments are accurately captured. Our primary objective is to filter out short-term fluctuations and statistical noise while retaining significant trends that extend beyond simple linear patterns. Achieving this requires a flexible method that does not impose overly restrictive structures, coupled with a robust approach capable of handling small sample sizes. For these reasons, we opted for P-splines, which allow for distinct smoothing parameters over age and time, enabling different degrees of smoothness across these domains[53,54]. This flexibility is particularly valuable in mortality analysis, where patterns can vary sharply by age and evolve either gradually or abruptly over time. P-splines also provide a robust and computationally efficient framework for handling noisy or sparse data, which is common in small-area estimation. To further speed up computations, an important consideration when dealing with a large number of regions, we used generalized linear array models (GLAM) framework, which enables fast evaluation of design matrices and penalty structures[55]. Since our analysis includes age 0, where mortality behaves very differently, we use a modified basis that partially decouples infant mortality from the rest of the age pattern[21].

A key challenge when working with very small sample areas is that objective criteria for selecting smoothing parameters often lead to excessive smoothing, potentially reducing complex trends to over-simplified linear approximations, particularly over time. To address this, we constrained the range of smoothing parameters and selected the optimal value within this range using the Bayesian information criterion (BIC). This approach serves as an implicit prior, effectively avoiding both over-smoothing (e.g., imposing linearity over time) and overfitting (e.g., directly interpolating observed data).

An additional advantage of the P-spline framework is its integration within the composite link model (CLM), which accommodates age-group structures frequently found in small-area mortality data. This approach allows us to simultaneously smooth age-time patterns and disaggregate grouped age data, yielding estimates for every single year of age. Given the large number of spatial units with consistent age-group structures over time, we adopted the computational method proposed by Camarda and Durbán[22], which significantly enhances computational efficiency for smoothing and ungrouping tasks.

To illustrate our methods, Supplementary Fig. 5 presents the trend in life expectancy at birth for 10 French and Dutch regions with varying profiles (geographical location, population size, life expectancy levels) for females from 1992 to 2019. The dots represent life expectancy levels calculated using raw data, while the lines represent life expectancy levels estimated using our methods, along with 95% confidence intervals.

All calculations were carried out using R version 4.4.2[56]. Routines used to compute the regional lifetables, as well as those used to generate all our figures, are openly accessible in an Open Source Framework repository, which provides detailed documentation to facilitate their use: https://osf.io/3dhmz. This repository also includes a full regional dataset containing annual population counts, life expectancy at birth, and year-to-year changes in life expectancy at birth by sex for the 450 regions in our panel.

### Reporting summary

Further information on research design is available in the Nature Portfolio Reporting Summary linked to this article.

## Data availability

Austria: Death counts are available at Statistik Austria: https://www.statistik.at/en/databases/statcube-statistical-database. Population counts are available at Eurostat: https://ec.europa.eu/eurostat/web/main/data/database. Belgium: Death and population counts are available at Eurostat: https://ec.europa.eu/eurostat/web/main/data/database. Denmark: Death and population counts are available at Statistics Denmark: https://www.statbank.dk. France: Death and population counts have been collected within the French Human Mortality Database project: https://frdata.org/en/french-humanmortality-database/. Germany: Death counts for German regions can be requested for a fee at the research data centre of the statistical offices of the German Länder: https://www.forschungsdatenzentrum.de/de/gesundheit/todesursachen. Population counts can be requested at the Federal Statistical office: https://www.destatis.de/EN/Service/Contact/_Contact.html. Ireland: Death and population counts are available at the Human Mortality Database: https://www.mortality.org/. Italy: Death and population counts are available at Istat: https://www.istat.it/en/population-and-households?data-and-indicators. Luxembourg: Death and population counts are available at the Human Mortality Database: https://www.mortality.org/. Netherlands: Death and population counts are available at Statistics Netherlands: https://opendata.cbs.nl/statline/#/CBS/nl/. Portugal: Death counts are available at Statistics Portugal: https://www.ine.pt/xportal/xmain?xpid=INE&xpgid=ine_base_dados&contexto=bd&selTab=tab2&xlang=en. Population counts are available at Eurostat: https://ec.europa.eu/eurostat/web/main/data/database. Spain: Death and population counts are available at the Spanish Statistical Office: https://www.ine.es/en/. Switzerland: Death and population counts are available at the Federal Statistical Office: https://www.pxweb.bfs.admin.ch/pxweb/en/. United Kingdom: Population counts for England and Wales are available at the Office for National Statistics: https://www.ons.gov.uk/peoplepopulationandcommunity/populationandmigration/populationestimates/datasets/. Death counts for England and Wales are

available at the Office for National Statistics: https://www.ons.gov.uk/peoplepopulationandcommunity/healthandsocialcare/causesofdeath/datasets/. Death counts for England and Wales from 1992 to 2001 were subject to a user-requested dataset. Death and population counts for Northern Ireland are available at the Northern Ireland Statistics and Research Agency: https://www.nisra.gov.uk/statistics. Deaths and population counts for Scotland are available at National Records of Scotland: https://www.nrscotland.gov.uk/statistics-and-data/The source of the shapefiles for all countries except Germany is Eurostat/GISCO. https://ec.europa.eu/eurostat/web/gisco/geodata/statistical-units/territorial-units-statistics©EuroGeographics for the administrative boundaries. We relied on the NUTS classification of 2021, except England and Wales, where the NUTS classification of 2016 was used. The original shapefiles for Italy were modified to match mortality data. The source of the shapefiles for Germany is the Federal Agency for Cartography and Geodesy (BKG). https://www.bkg.bund.de/EN/Home/home.html.©GeoBasis-DE (BKG). Detailed values obtained for our 450 regions are available at: https://osf.io/3dhmz/. Source data are provided with this paper.

## Code availability
Routines to replicate our lifetables in the case of French and Austrian regions, as well as codes to replicate all the Figures and Tables are available at: https://osf.io/3dhmz/.

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

## Acknowledgements
This work was supported by funding from the European Research Council (ERC) under the European Union's Horizon 2020 research and innovation programme (grant agreement No 851485).

## Author contributions
F.B., P.G., C.G.C. and F.M. conceptualized the study. C.G.C. and F.B. developed the methodology. I.A., P.G., F.B and M.M. collected and curated the data. F.B., P.G., C.G.C. and F.M. performed the formal analysis. P.G. and F.B. drafted the original manuscript. P.G., F.B., S.K., C.G.C., J.T., M.M., F.M. and I.A. reviewed and edited the manuscript. F.B. and P.G. produced the visualizations.

## Funding

## Competing interests
The authors declare no competing interests.
