## [Transparent Peer Review file · Nature Communications]

Potential and challenges for sustainable progress in human longevity

Corresponding Author: Dr Pavel Grigoriev

Version 0:

Reviewer comments:

Reviewer #1

(Remarks to the Author)

1. Key Results

This article presents a comprehensive analysis of subnational trends in life expectancy across 450 regions in Western Europe between 1992 and 2019. The authors identify a period of convergence in mortality trends up to 2005, followed by a marked deceleration and divergence. This divergence is primarily due to a sharper slowdown in life expectancy improvements among laggard regions compared to vanguard regions, which maintained a more stable pace of progress. The paper shows that the stagnation is largely attributable to increasing mortality or slowed progress in the 55–74 age group. This work contributes valuable insights into the geographic dynamics of longevity, highlighting both potential limits and windows for further gains.

2. Validity

The empirical analysis is generally sound and based on well-established demographic techniques. However, several methodological and interpretative issues warrant clarification:

- The use of averages across regions may mask substantial intra-regional social inequalities in mortality. These internal disparities may affect both the observed means and variances and deserve at least a brief acknowledgment in the limitations. Notably, recent work has documented the persistence and transformation of social mortality gradients across Europe, both in the long run (Luque de Haro, 2024) and in recent decades (Mackenbach et al., 2018), which makes it crucial to consider socioeconomic variation within regions as a potentially confounding or explanatory factor.
- The paper would benefit from a more explicit justification of the three biennia selected (1992–93, 2004–05, and 2018–19). While it appears that the choice of 2004–05 is intended to reflect a turning point in the trend of life expectancy gains, this rationale should be stated more explicitly. Additionally, this biennium does not correspond to the exact midpoint of the observation period (which would fall closer to 2005–06). Clarifying this choice is especially relevant given that the right-hand panels of Figures 4 and 5 — which summarize decile-level transformations in life expectancy rankings — reflect cumulative changes over the entire period, not just within these biennia.
- In Figures 4 and 5, the left-hand panels illustrate the pace of life expectancy improvements during the selected biennia. While this snapshot is informative for assessing whether top-ranked regions are maintaining or accelerating their lead, it would be valuable to complement it with an additional visualization showing cumulative gains in life expectancy over broader subperiods (e.g., 1992–2005 and 2005–2019) for each territorial unit. Such a depiction would enhance the reader's ability to interpret long-term regional dynamics and evaluate convergence/divergence trajectories over time.
- The assertion "as expected" (p. 9, l. 188) is not sufficiently supported by a prior argument or theoretical expectation, and should either be justified or rephrased.

3. Significance

The manuscript addresses an important issue of broad interest in demography and public health: the evolving potential for further gains in life expectancy in high-income settings. While the empirical contribution is strong, the paper would benefit from a richer theoretical contextualization. In particular, the discussion could be strengthened by connecting the findings to one of the classical debates in historical demography (notably the McKeown thesis, Szreter's critiques, and the role of behavioral and institutional factors in the mortality decline).

Furthermore, the discussion could benefit from a consideration of countervailing mechanisms, that is, forces or processes that may undermine or reverse the expected trajectory of health improvements, even in regions with relatively favorable socioeconomic and institutional conditions. This notion has been acknowledged in the literature on socioeconomic status and health disparities, for instance by Lutfey and Freese (2005), who note the possibility that certain mechanisms may disadvantage higher-status groups under specific circumstances.

In the context of this study, such countervailing mechanisms may include rising exposure to environmental risks (e.g., pollution, heat stress), the diffusion of unhealthy consumption patterns, or emerging inequalities in health behaviors (tobacco alcohol and other drugs consumption) and chronic disease management. A recent study by Masselot et al. (2023) has shown that climate-related factors, particularly heatwaves, are already contributing significantly to excess mortality across European cities. Acknowledging these emerging threats would provide a more nuanced interpretation of why some regions experience stagnation or reversal in longevity trends despite apparent structural progress.

4. Data and Methodology

- The spatial disaggregation used is a major strength comparing with other national studies. Nevertheless, the variation in population sizes across regions may influence both mean life expectancy and its variance (e.g., smaller units tend to exhibit more variability). A supplementary table indicating the population size of each unit at different time points would be a valuable addition.

- The use of NUTS-based spatial units, while standard in European demographic research, results in considerable variation in population size across countries and regions. For instance, NUTS-3 regions in Spain or Italy differ markedly from NUTS-2 regions used in Portugal or Belgium, and from NUTS-0 units in Ireland or Luxembourg. These discrepancies may affect both the internal heterogeneity of units and the statistical stability of mortality estimates. A brief discussion of how these differences were addressed or accounted for would improve the transparency of the methodology. For example, even within the NUTS-3 level, there are stark contrasts: the Madrid region is a large, socioeconomically diverse metropolitan area with over six million inhabitants, while Teruel is sparsely populated, more homogeneous, and has fewer than 150,000 inhabitants. Such differences are likely to affect both variability and internal inequality, and merit consideration.

- While the structure of the journal often places methods at the end, a brief summary of the main data sources and modeling strategy in the Introduction would aid clarity for readers.

- It may be useful to include in the appendix the figure currently only available online showing all regional deciles, which would complement the information in Figures 4 and 5.

5. Analytical Approach

- The use of P-splines and CLM for small-area mortality estimation appears methodologically appropriate, but the details could be better summarized in the main text.

- It would be helpful if the authors clarified how they validated the smoothing parameters and whether they tested alternative specifications or sensitivity to different degrees of smoothing.

- The indicators chosen (e_0 , 20q35, 20q55, 10q75) are relevant and well-established, although the discussion could explore whether the trends observed in 20q55 may also be influenced by alcohol- and diet-related mortality, not just smoking.

- To improve transparency, the authors might consider including a supplementary figure comparing raw and smoothed life expectancy trajectories for a small number of representative regions. This would help readers better understand the impact of the smoothing procedure and assess whether key patterns are robust to different levels of smoothing. Such a visualization would complement the existing methodological description and strengthen the overall clarity of the analytical approach.

6. Suggested Improvements

- Consider incorporating a broader discussion of behavioral risk factors beyond smoking (e.g., alcohol consumption, diet), especially considering the age group most affected by the stagnation (55–74).

- Explore, if possible, common structural features of laggard and vanguard regions (e.g., education levels, income inequality, per capita income, health service access, productive structure), which could enrich the interpretation of long-term regional dynamics.

- If feasible, include a brief note acknowledging that even subnational units such as provinces (e.g., in Spain) may still mask significant internal disparities, and that this limitation could affect the interpretation of regional averages.

- The authors state that “on average, these ‘new’ vanguard regions experienced absolute life expectancy gains comparable to those of the original vanguard regions.” While this is technically accurate, it may obscure an important dynamic. For a region to move into the top decile of life expectancy, it must have experienced faster gains than the original vanguards over a substantial part of the observation period. Therefore, the fact that these new vanguard regions are now improving at the same pace as the original ones suggests a relative deceleration in their trajectory. A brief clarification of this point would enhance the interpretation of regional dynamics and the discussion of convergence/divergence processes.

7. Clarity and Context

- The manuscript is generally well-written and accessible, but could benefit from clearer theoretical grounding, as noted above.

- Adding more context about long-term historical trends in mortality and prior health transitions would help frame the results more robustly.

8. References

- The manuscript cites relevant and up-to-date sources. However, additional references to key debates in demographic history and epidemiology (e.g., McKeown, Szreter, Omran’s transition theory) would enhance the contextual depth of the discussion.

9. Expertise and Scope

- I am confident assessing the demographic interpretation of mortality trends, regional analysis, and socioeconomic contextualization.

- I do not have deep expertise in the specific statistical implementation of P-splines or CLM, and thus cannot fully evaluate

the appropriateness of the smoothing parameter selection beyond what is reported.

References

- Luque de Haro, V. A. (2024). Social inequalities in adult mortality across Europe (18th–21st centuries): A critical analysis of theories and evidence. *Economics & Human Biology*, 55. <https://doi.org/10.1016/j.ehb.2024.101438>
- Lutfey, K., & Freese, J. (2005). Toward some fundamentals of fundamental causality: Socioeconomic status and health in the routine clinic visit for diabetes. *American Journal of Sociology*, 110(5), 1326–1372. <https://doi.org/10.1086/428914>
- Mackenbach, J. P., Valverde, J. R., Artnik, B., et al. (2018). Trends in health inequalities in 27 European countries. *Proceedings of the National Academy of Sciences*, 115(25), 6440–6445. <https://doi.org/10.1073/pnas.1800028115>
- Masselot, P., Mistry, M., Vanoli, J., et al. (2023). Excess mortality attributed to heat and cold: a health impact assessment study in 854 cities in Europe. *The Lancet Planetary Health*, 7(4), e271–e281. [https://doi.org/10.1016/S2542-5196\(23\)00023-2](https://doi.org/10.1016/S2542-5196(23)00023-2)

(Remarks on code availability)

Reviewer #2

(Remarks to the Author)
Summary of Key Results

This manuscript examines changes in life expectancy from 1992 to 2019 across Western European regions. The principal strength of the study lies in the diversity of regions analyzed simultaneously, which may hold important implications for public policy if interpreted appropriately. The authors use these regions to rank gains in life expectancy and classify them as either laggards or vanguards in life expectancy improvements. The results indicate a slowdown in the pace of mortality reductions during the second half of the study period. Furthermore, the authors investigate the drivers of this deceleration in gains in e_0 , identifying the 55–74 age group as having the greatest impact on the reduction in gains.

Significance and Impact

The findings may hold considerable relevance for policymakers and researchers examining the deceleration in life expectancy improvements. The analysis is insightful, illustrating the value of considering both local and broader contextual factors when evaluating the limits of longevity.

Data and Methodology

The study is fully reproducible (with the exception of the dataset, which is not fully publicly available; however, this is not the responsibility of the authors). The documentation for code usage is generally thorough and accessible, except the file “0-Functions-R,” which would benefit from more detailed documentation regarding the purpose and function of each component. The code is well-structured and clear. A minor technical note is that the analysis relies on the MortalitySmooth library, which has been deprecated on CRAN. Since this requires additional steps for installation in R, it would be helpful to provide users with brief guidance on how to complete this process.

Two methodological concerns are not addressed in the manuscript and warrant discussion. First, the issue of mixing NUTS classification levels. The NUTS classification distinguishes between NUTS-1 (major socio-economic regions), NUTS-2 (basic regions for regional policies), and NUTS-3 (small regions for specific diagnoses). The authors do not clarify why it is methodologically appropriate to compare, for example, Spain at the NUTS-3 level with Portugal at the NUTS-2 level. This inconsistency may obscure heterogeneities that the paper aims to uncover. What changes, if any, arise when the analysis is restricted to the NUTS-2 level?

Second, the use of P-splines raises concerns. Although commonly used and well-documented in demographic research, P-splines are known to perform poorly at the extremes of the age distribution, particularly at ages 0 and w . Given the aims of the paper, overfitting at age 0 may be especially problematic, as errors at this age can propagate through the life table and significantly affect estimates of e_0 . The code suggests that some correction was applied for age 0, but this is not explained in the methods or the appendix. The authors are encouraged to elaborate on how or if this issue was addressed.

Analytical Approach

The methods employed are appropriate for addressing the research question. They are sufficiently comprehensive, and the discussion of results is both adequate and original.

Suggested Improvements

In the abstract, the authors state: “... is strongly associated with mortality of the population aged 55–74, which increased by 2019...”. It appears the intention is to refer to the population rather than mortality increasing; the sentence is currently ambiguous.

The rationale for using regional estimates of e_0 and comparing them should be emphasized more clearly early in the manuscript. Although the importance of this approach becomes clearer in the results and discussion sections, it would benefit from stronger justification earlier on.

The authors explain why the analysis concludes in 2019, but do not justify why it begins in 1992. This should be addressed. Although two distinct periods of e_0 gains are identified in Figure 1, these classifications are not applied in the subsequent analysis. Incorporating them may enhance the strength of the findings.

For Figure 3, the color scale should be modified to ensure readability when printed in black and white.

On page 6, line 162, the authors state: “...today’s laggard regions are significantly lower than...”. It should be clarified

whether this refers to statistical significance.

In the Shiny app, the figures lack a color legend, which should be included.

On page 12, line 273, the sentence "... phenomenon has only been observed in the United Kingdom..." should be nuanced to reflect that this applies only among the countries included in the study.

Questions for the authors:

- Given that mortality at younger ages is relatively low in most of the regions analyzed, some may argue that other indicators, such as LE25 or LE65, may be more suitable for assessing changes in old-age mortality. Why were these not considered?
- Since the analysis aims to identify the age group contributing most to the stagnation in life expectancy gains, why was an age decomposition not performed?
- P-splines allow for the estimation of confidence intervals through bootstrapping. Would incorporating confidence intervals into the e0 estimates add value to the analysis, particularly in assessing whether regional and national differences are statistically significant?
- In the discussion of future research, the authors suggest incorporating COVID-19 years. Why not include them in the current analysis?

(Remarks on code availability)

Reviewer #3

(Remarks to the Author)

The paper presents a careful analysis of mortality and life expectancy trends across 450 subnational regions in Western Europe over a nearly 30-year period. It revealed sustained improvements in life expectancy in vanguard regions, while demonstrating different dynamics in laggard areas. This is an original, methodologically sound research, supported by beautiful visualization.

A few comments: `

- I wonder whether the Introduction could be slightly extended or adjusted to better reflect the broader context of diverse trajectories in life expectancy across vanguard and laggard subpopulations.
- In the Discussion, you refer to convergence-divergence theory and suggest that divergence may result from faster growth in vanguard regions due to earlier adoption of innovations. However, based on your results, divergence since the mid-2000s appears to be more strongly driven by a slowdown in laggard regions. It would be useful to clarify this point and consider whether the convergence-divergence framework fully captures the observed pattern.
- While the manuscript is descriptively rich, a more detailed discussion of the potential causal mechanisms behind the recent divergences would be beneficial. Other factors, such as differential access to healthcare, austerity measures, or economic restructuring, may be relevant beyond smoking and cardiovascular disease. These factors could be acknowledged, even if they are speculative.
- Although the focus on the pre-pandemic period is justified, I suggest briefly commenting on how COVID-19 might have affected the trajectories described. Even a short reflection on the potential for COVID-19 to exacerbate or alter divergence patterns would be a useful addition.

Minor comments:

- I recommend replacing "during the pre-pandemic period" with the actual years ("1992–2019"). Consider using "slowdown" or similar alternatives instead of "deceleration" (in some cases). I would also advise avoiding strong phrases like "we strongly advocate."
- The paper opens with references to COVID-19, which is not the focus of the study. It might make more sense to start with literature on pre-pandemic stagnation in mortality improvements and only later mention the pandemic if needed.
- The following might be a useful addition to the discussion of stagnating mortality improvements in other high-income countries:
Timonin, S., Leon, D. A., Banks, E., Adair, T., & Canudas-Romo, V. (2024). Faltering mortality improvements at young-middle ages in high-income English-speaking countries. *International Journal of Epidemiology*, 53(3).
<https://doi.org/10.1093/ije/dyae128>
- Should confidence intervals be shown, at least for regions with maximum and minimum life expectancy? Also, I couldn't find a legend explaining the meaning of the colored lines. Using different Y-axes for males and females makes comparisons difficult and may be misleading.
- Typically, D1 refers to the most deprived group and D10 to the least deprived (e.g., in the UK and Australia).

(Remarks on code availability)

Version 2:

Reviewer comments:

Reviewer #1

(Remarks to the Author)

The authors have carefully and thoroughly addressed all my previous comments. The revised manuscript shows substantial improvement in theoretical contextualization, methodological transparency, and clarity of presentation. The additional supplementary materials (Figures 1 and 5, Tables 1–2) meaningfully enhance interpretability and reproducibility. I find the paper suitable for publication in its current form.

Minor issue

Please correct the bibliographic entry for Luque de Haro (2024) in the reference list.

The compound surname is incorrectly formatted as “De Haro,” and the journal title should include the ampersand symbol.

Correct form:

Luque de Haro, V. A. (2024). Social inequalities in adult mortality across Europe (18th–21st centuries): A critical analysis of theories and evidence. *Economics & Human Biology*, 55, 101438. <https://doi.org/10.1016/j.ehb.2024.101438>

(Remarks on code availability)

Reviewer #2

(Remarks to the Author)

The authors have done an excellent job addressing all previous comments and substantially improving the manuscript. The revised version is clear, comprehensive, and much stronger. The discussion has been expanded in meaningful ways, providing a richer theoretical context and situating the findings within broader demographic debates. All my earlier concerns have been carefully considered and satisfactorily resolved. In my opinion, the manuscript is now ready for publication. Congratulations to the authors!

(Remarks on code availability)

I have reviewed the revised version of the code. The authors have addressed my previous concerns thoroughly, providing clearer documentation and improving accessibility. The updated code is now well-structured and fully reproducible. I am satisfied with these revisions and appreciate the authors' careful attention to detail.

Reviewer #3

(Remarks to the Author)

Many thanks for carefully addressing the comments of all the reviewers, including mine. I have no further suggestions.

(Remarks on code availability)

Authors' response

We are very grateful to the reviewers for their constructive comments and suggestions, which helped us to improve the manuscript. Below we provide our point-by-point responses in blue.

REVIEWER COMMENTS

Reviewer #1 (Remarks to the Author):

1. Key Results

This article presents a comprehensive analysis of subnational trends in life expectancy across 450 regions in Western Europe between 1992 and 2019. The authors identify a period of convergence in mortality trends up to 2005, followed by a marked deceleration and divergence. This divergence is primarily due to a sharper slowdown in life expectancy improvements among laggard regions compared to vanguard regions, which maintained a more stable pace of progress. The paper shows that the stagnation is largely attributable to increasing mortality or slowed progress in the 55–74 age group. This work contributes valuable insights into the geographic dynamics of longevity, highlighting both potential limits and windows for further gains.

2. Validity

The empirical analysis is generally sound and based on well-established demographic techniques. However, several methodological and interpretative issues warrant clarification:

The use of averages across regions may mask substantial intra-regional social inequalities in mortality. These internal disparities may affect both the observed means and variances and deserve at least a brief acknowledgment in the limitations. Notably, recent work has documented the persistence and transformation of social mortality gradients across Europe, both in the long run (Luque de Haro, 2024) and in recent decades (Mackenbach et al., 2018), which makes it crucial to consider socioeconomic variation within regions as a potentially confounding or explanatory factor.

Following the suggestion of the Reviewer, we now briefly acknowledge this limitation (Line 350):

“Moreover, while we aimed for the most detailed geographical units and the longest possible time series, differences in data availability obliged us to use less detailed information in some countries. This averaging of outcomes across larger areas may conceal inequalities in longevity, which recent studies have shown to be important for interpreting spatial disparities [47, 48].”

The paper would benefit from a more explicit justification of the three biennia selected (1992–93, 2004–05, and 2018–19). While it appears that the choice of 2004–05 is intended to reflect a turning point in the trend of life expectancy gains, this rationale should be stated more explicitly. Additionally, this biennium does not correspond to the exact midpoint of the observation period (which would fall closer to 2005–06). Clarifying this choice is especially

relevant given that the right-hand panels of Figures 4 and 5 — which summarize decile-level transformations in life expectancy rankings — reflect cumulative changes over the entire period, not just within these biennia.

Our paper focuses on the analysis of long-term mortality trends for the period 1992-2019. The initial point of the analysis (1992) corresponds to the earliest year for which the data were available for all countries included in our sample. The year 2019 corresponds to the last year before the COVID-19 pandemic. Indeed, 2004-2005 reflects a turning point in the trend of life expectancy gains. This point also falls very close to the mid-point of the entire period. As a result, we ended up with three snapshots: 1992-1993, 2004-2005, and 2018-2019. Although the respective clarification was provided in our initial submission (p.4, lines 112-116), we agree that more explanations are needed.

The first sentence in the 'Results' section (Line 72) reads now as follows:

"Figure 1 offers a dual perspective on annual gains in life expectancy at birth (hereafter life expectancy or e_0) across 450 regions in Western Europe from 1992 (the earliest year for which data are available for all of the countries included) to 2019 (the last year before the COVID-19 pandemic)."

We would like to clarify that the purpose of Figure 1 was not only to provide the global overview of the changes in life expectancy gains but also to justify the usage of 2004-2005 as one of the three snapshots over the observation period. These snapshots were then used consistently for Figures 3–6.

In Figures 4 and 5, the left-hand panels illustrate the pace of life expectancy improvements during the selected biennia. While this snapshot is informative for assessing whether top-ranked regions are maintaining or accelerating their lead, it would be valuable to complement it with an additional visualization showing cumulative gains in life expectancy over broader subperiods (e.g., 1992–2005 and 2005–2019) for each territorial unit. Such a depiction would enhance the reader's ability to interpret long-term regional dynamics and evaluate convergence/divergence trajectories over time.

We agree with this suggestion. We added Supplementary Figure 1 to Supplementary Information.

The assertion 'as expected' (p. 9, l. 188) is not sufficiently supported by a prior argument or theoretical expectation, and should either be justified or rephrased.

'as expected' has been removed from this sentence.

3. Significance

The manuscript addresses an important issue of broad interest in demography and public health: the evolving potential for further gains in life expectancy in high-income settings. While the empirical contribution is strong, the paper would benefit from a richer theoretical contextualization. In particular, the discussion could be strengthened by connecting the

findings to one of the classical debates in historical demography (notably the McKeown thesis, Szreter's critiques, and the role of behavioral and institutional factors in the mortality decline).

Thank you for these very useful suggestions. We have added the following paragraphs to the 'Discussion' section (Line 306):

“Although our analyses are mainly aimed at providing novel empirical insights, and a detailed analysis of mortality determinants is beyond the scope of this study, it is important to highlight factors and potential causal mechanisms behind the observed subnational trends and differentials. Broadly speaking, it makes sense to situate the discussion of these factors within long-standing debates on the role of broad socioeconomic change versus targeted interventions in shaping mortality decline [35, 36]. The classical McKeown thesis attributes rapid mortality decline in industrialized countries to broad economic and social changes, rather than to targeted public health or medical interventions. While this controversial thesis has since been overturned on empirical grounds, it posed the fundamental question of whether population health is determined primarily by broad-based social, political and economic conditions or by more targeted policies and interventions [37].

At the subnational level, this broader debate finds a close parallel in discussions of the relative role of individual versus contextual determinants of health [38]. Accordingly, regional disparities in mortality can be interpreted in terms of compositional effects (reflecting the unequal distribution of individual risk factors across areas) or contextual effects (the influence of place-based conditions and policies). From a compositional perspective, our finding that stagnation in life expectancy has been most pronounced at ages 55–74 highlights the contribution of lifestyle-related factors such as alcohol consumption, drug use, poor diet and physical inactivity [1, 39]. At the same time, contextual factors provide important complementary explanations. Recent research has emphasized the impact of austerity measures and so-called deaths of despair (suicide, drug misuse and alcohol-specific causes) on stagnating mortality in the UK [40–42]. In our study, the regional divergence in life expectancy observed since the mid-2000s broadly coincides with the period of the 2008 financial crisis and its aftermath, when economic growth slowed across Western Europe while becoming increasingly concentrated in major metropolitan regions [43, 44]. These urban areas, which host high-skill service industries, are now among the best-performing regions in terms of longevity. For instance, between 2005 and 2019, the western part of central London rose from 132nd to 1st place for males and from 247th to 15th for females, while Paris advanced from 40th to 32nd for males and from 45th to 4th for females.”

Furthermore, the discussion could benefit from a consideration of countervailing mechanisms, that is, forces or processes that may undermine or reverse the expected trajectory of health improvements, even in regions with relatively favorable socioeconomic and institutional conditions. This notion has been acknowledged in the literature on socioeconomic status and health disparities, for instance by Lutfey and Freese (2005), who note the possibility that certain mechanisms may disadvantage higher-status groups under specific circumstances. In

the context of this study, such countervailing mechanisms may include rising exposure to environmental risks (e.g., pollution, heat stress), the diffusion of unhealthy consumption patterns, or emerging inequalities in health behaviors (tobacco alcohol and other drugs consumption) and chronic disease management. A recent study by Masselot et al. (2023) has shown that climate-related factors, particularly heatwaves, are already contributing significantly to excess mortality across European cities. Acknowledging these emerging threats would provide a more nuanced interpretation of why some regions experience stagnation or reversal in longevity trends despite apparent structural progress.

This is a very good suggestion, thank you. The issues mentioned are indeed among the challenges implied by the title of the paper and should therefore be emphasized. We added the following paragraph in the discussion (Line 333):

“Looking ahead, emerging public health threats could slow or even reverse longevity improvements, even in regions with favourable socioeconomic conditions. Previous research has suggested that countervailing mechanisms such as increased exposure to environmental risks and the diffusion of unhealthy behaviours may disproportionately affect higher-status groups [45]. Climate-related risks, notably heatwaves, are already having a measurable impact on mortality in European cities, with particularly strong effects observed in eastern Europe [46]. 2023). These factors are likely to become increasingly influential in the future, although well-designed policy interventions could mitigate their adverse effects on population health.”

4. Data and Methodology

The spatial disaggregation used is a major strength comparing with other national studies. Nevertheless, the variation in population sizes across regions may influence both mean life expectancy and its variance (e.g., smaller units tend to exhibit more variability). A supplementary table indicating the population size of each unit at different time points would be a valuable addition.

Following your suggestion, we have added in our Open Science Framework repository a full regional dataset that contains annual population counts, life expectancy at birth, and year-to-year changes in life expectancy at birth by sex for the 450 regions in our panel indicating the population size of each unit at different time points. We have mentioned this dataset at the end of the Methodology section (Line 455).

“This repository also includes a full regional dataset containing annual population counts, life expectancy at birth, and year-to-year changes in life expectancy at birth by sex for the 450 regions in our panel.”

The use of NUTS-based spatial units, while standard in European demographic research, results in considerable variation in population size across countries and regions. For instance, NUTS-3 regions in Spain or Italy differ markedly from NUTS-2 regions used in Portugal or Belgium, and from NUTS-0 units in Ireland or Luxembourg. These discrepancies may affect both the internal heterogeneity of units and the statistical stability of mortality estimates. A

brief discussion of how these differences were addressed or accounted for would improve the transparency of the methodology. For example, even within the NUTS-3 level, there are stark contrasts: the Madrid region is a large, socioeconomically diverse metropolitan area with over six million inhabitants, while Teruel is sparsely populated, more homogeneous, and has fewer than 150,000 inhabitants. Such differences are likely to affect both variability and internal inequality, and merit consideration.

You raised legitimate concerns about the comparability of mortality estimates between different spatial units. We must emphasize that the selection of spatial units for our analysis was driven by data availability. Demographic statistics are typically tabulated according to established administrative units, which align with the NUTS classification. Although the NUTS classification was created to maintain comparable regional units across the EU, substantial comparability issues could arise due to variations in population size, population density, and administrative structures of the regions at the same NUTS level. Therefore, the methodological issues you highlighted are mainly related to the limitations of the NUTS classification per se rather than the design of our study. As mentioned above, there was no other alternative for us as to use the regional mortality data available at different NUTS levels.

There was considerable variation among the European countries in terms of data availability. For some countries, the detailed regional mortality data are readily available online, while for others such data can only be obtained upon special request. In certain cases, the desired level of geographical details is not possible to obtain because of data protection rules. While building our dataset we aimed at the highest level of geographical detail, the longest possible time series, and the largest possible number of countries. We also kept in mind data comparability issues and the suitability of the data for visualizations, especially mortality maps. Although we believe that we ended with a decent data sample, we agree that the comparability between spatial units warrant thorough discussion. Thus, we added the following paragraph under the 'Strengths and limitations' section (Line 342).

“A major strength of our work lies in the unique combination of wide geographical coverage with high spatial resolution mortality series. We invested substantial effort to harmonize mortality data across regions and over time, ensuring comparability despite differences in national statistical practices (see Supplementary Table 3 for details on the implemented adjustments). This harmonization was essential to meeting our study’s central objective: assessing long-term longevity trends both by individual region and by contrasting vanguard and laggard groups. Nonetheless, perfect comparability could not be achieved. The use of the NUTS classification, though standard in European demographic research, entails considerable variation in the population size of spatial units within the same NUTS level between countries. Moreover, while we aimed for the most detailed geographical units and the longest possible time series, differences in data availability obliged us to use less detailed information in some countries. This averaging of outcomes across larger areas may conceal inequalities in longevity, which recent studies have shown to be important for interpreting spatial disparities [47, 48].”

While the structure of the journal often places methods at the end, a brief summary of the main data sources and modeling strategy in the Introduction would aid clarity for readers.

As suggested, we added a brief summary of the main data sources and modelling strategy to the Introduction (Line 45). The revised text reads as follows:

“Another advantage of Western Europe is that detailed and reliable long-term mortality data stratified by year, region, sex, and age are generally available. Nevertheless, the data routinely collected by national statistical offices are far from homogeneous and cannot be directly used for comparative analyses (Supplementary Table 3). We thus begin by harmonizing the data to ensure a uniform structure and the comparability of spatial units over the observation period. We then apply a state-of-the-art demographic methodology [21, 22] to smooth the data and calculate mortality indicators.”

It may be useful to include in the appendix the figure currently only available online showing all regional deciles, which would complement the information in Figures 4 and 5.

We assume that the reviewer refers to the maps available in our Shiny App, which display the rank of each region in the life expectancy distribution by each year and sex. In the article, we chose to present such maps only for three benchmark periods, and only for the vanguard and laggard regions. We would prefer not to overwhelm the reader with excessive additional information. Instead, we provide the opportunity to explore our results in full using the Shiny App, which we have further improved.

https://histdemo.shinyapps.io/ReLoG_Europe/

5. Analytical Approach

The use of P-splines and CLM for small-area mortality estimation appears methodologically appropriate, but the details could be better summarized in the main text.

We have expanded the Methodology section to better summarize the rationale behind the use of P-splines in our context. Specifically, we now emphasize their flexibility in allowing different smoothing parameters over age and time, which is particularly valuable given the distinct mortality patterns across ages (e.g., infant vs. adult mortality) and their evolution over time. We also highlight that P-splines provide a robust and efficient approach for dealing with sparse or noisy data, which is common in small-area mortality estimation. Finally, we clarify that the use of the Generalized Linear Array Models (GLAM) framework allows for faster computation, which is especially important when analyzing a large number of regions. We hope this revision (and the associated added references) improves the clarity and accessibility of our methodological choices.

It would be helpful if the authors clarified how they validated the smoothing parameters and whether they tested alternative specifications or sensitivity to different degrees of smoothing.

Thank you for the insightful comment regarding the validation of the smoothing parameters and the potential sensitivity of our results to different degrees of smoothing. In our analysis,

smoothing parameters - controlling the amount of smoothness in the P-spline framework - are selected using the Bayesian Information Criterion (BIC). This criterion balances model fit and complexity, where fit is measured by the Deviance in a Poisson setting (McCullagh & Nelder, 1989, p. 34), and complexity is quantified through the effective dimension, computed as the trace of the hat matrix (Hastie & Tibshirani, 1990, p. 52). A key advantage of P-splines is that both these quantities are computed analytically from the estimation algorithm. BIC, in particular, has been shown to perform well in the context of mortality modelling, often favouring more parsimonious models than criteria such as AIC (Currie et al., 2004; Camarda, 2012).

In dealing with regional data, we encountered significant heterogeneity: small regions often exhibit high variability due to sparse populations, while larger regions provide more stable information. Rather than imposing a fixed degree of smoothing, we allowed the smoothing parameters to be selected by BIC, enabling the level of smoothness to adapt to the data characteristics of each region.

A further strength of our approach lies in the anisotropic nature of P-splines, which allows separate tuning of smoothness over age and time. We permitted a wide range of values for the smoothing parameter over age, while imposing a mild prior on the parameter over time to avoid extremes, namely, overly small values that could lead to overfitting, and overly large values that would flatten trends excessively. Even so, the candidate values for the time-dimension smoothing spanned a broad interval (from 10^{-2} to 10^2), ensuring substantial flexibility: values smaller than 10^{-2} would lead to practically interpolating mortality over time, whereas values larger than 10^2 would practically reduce the fit to a linear time trend at each age. BIC then selects the optimal smoothing level within this range independently for each region.

We hope this additional clarification and evidence help convey the rationale and robustness of our methodology.

The indicators chosen (e_0 , 20q35, 20q55, 10q75) are relevant and well-established, although the discussion could explore whether the trends observed in 20q55 may also be influenced by alcohol- and diet-related mortality, not just smoking.

Please see our response to the similar point below (6. Suggested Improvements).

Explore, if possible, common structural features of laggard and vanguard regions (e.g., education levels, income inequality, per capita income, health service access, productive structure), which could enrich the interpretation of long-term regional dynamics.

We are afraid that this would be too ambitious to accomplish within one study. However, we do agree that this would enrich the interpretation of long-term regional dynamics, and thus, we mention in the discussion (Line 379) that “identifying common characteristics (e.g. socioeconomic or cultural factors) that differentiate vanguard from laggard regions could shed more light on the reasons why some areas perform better than others. Addressing this

question would eventually require systematic analyses that integrate mortality outcomes with area-level contextual variables. While this lies beyond the scope of our study, it represents a promising and necessary direction for future research”.

To improve transparency, the authors might consider including a supplementary figure comparing raw and smoothed life expectancy trajectories for a small number of representative regions. This would help readers better understand the impact of the smoothing procedure and assess whether key patterns are robust to different levels of smoothing. Such a visualization would complement the existing methodological description and strengthen the overall clarity of the analytical approach.

We appreciate the reviewer’s suggestion to improve transparency by visually comparing raw and smoothed life expectancy trajectories for a small number of regions.

In line with these recommendations, we include a new Supplementary Figure 5. This figure shows the observed and estimated life expectancy at birth (with 95% confidence intervals) for 10 different French and Dutch regions, separately for males and females (based on publicly available data). These regions present large differences in terms of population size, location and level of life expectancy. The figure demonstrates that our BIC-driven procedure captures a wide array of regional time trends without imposing a common structure. Using alternative information criteria would likely have resulted in different smoothing selections that may not have captured the observed patterns as accurately, potentially leading to less reliable conclusions in subsequent analyses. We have added a paragraph at the end of the Methodology section (Line 447) to describe this new Figure:

“To illustrate our methods, Supplementary Figure 5 presents the trend in life expectancy at birth for 10 French and Dutch regions with varying profiles (geographical location, population size, life expectancy levels) for female from 1992 to 2019. The dots represent life expectancy levels calculated using raw data, while the lines represent life expectancy levels estimated using our methods, along with 95% confidence intervals.”

To facilitate transparency and reproducibility, we have also updated our OSF repository to include fully reproducible code for this additional analysis.

6. Suggested Improvements

Consider incorporating a broader discussion of behavioral risk factors beyond smoking (e.g., alcohol consumption, diet), especially considering the age group most affected by the stagnation (55–74).

Such discussion is now incorporated in the following paragraph (Line 316):

“At the subnational level, this broader debate finds a close parallel in discussions of the relative role of individual versus contextual determinants of health [38]. Accordingly, regional disparities in mortality can be interpreted in terms of compositional effects (reflecting the unequal distribution of individual risk factors across areas) or contextual effects (the influence

of place-based conditions and policies). From a compositional perspective, our finding that stagnation in life expectancy has been most pronounced at ages 55–74 highlights the contribution of lifestyle-related factors such as alcohol consumption, drug use, poor diet and physical inactivity [1, 39]. At the same time, contextual factors provide important complementary explanations. Recent research has emphasized the impact of austerity measures and so-called deaths of despair (suicide, drug misuse and alcohol-specific causes) on stagnating mortality in the UK [40–42]. In our study, the regional divergence in life expectancy observed since the mid-2000s broadly coincides with the period of the 2008 financial crisis and its aftermath, when economic growth slowed across Western Europe while becoming increasingly concentrated in major metropolitan regions [43, 44]. These urban areas, which host high-skill service industries, are now among the best-performing regions in terms of longevity. For instance, between 2005 and 2019, the western part of central London rose from 132nd to 1st place for males and from 247th to 15th for females, while Paris advanced from 40th to 32nd for males and from 45th to 4th for females.”

If feasible, include a brief note acknowledging that even subnational units such as provinces (e.g., in Spain) may still mask significant internal disparities, and that this limitation could affect the interpretation of regional averages.

We added the following statements to the ‘Strength and limitation’ section (Line 350):

“Moreover, while we aimed for the most detailed geographical units and the longest possible time series, differences in data availability obliged us to use less detailed information in some countries. This averaging of outcomes across larger areas may conceal inequalities in longevity, which recent studies have shown to be important for interpreting spatial disparities [47, 48].”

The authors state that 'on average, these ‘new’ vanguard regions experienced absolute life expectancy gains comparable to those of the original vanguard regions.' While this is technically accurate, it may obscure an important dynamic. For a region to move into the top decile of life expectancy, it must have experienced faster gains than the original vanguards over a substantial part of the observation period. Therefore, the fact that these new vanguard regions are now improving at the same pace as the original ones suggests a relative deceleration in their trajectory. A brief clarification of this point would enhance the interpretation of regional dynamics and the discussion of convergence/divergence processes.

This sentence is more linked to our perception of progress in human longevity. In a sense, our approach is in line with Oeppen and Vaupel (2002), as well as Vallin & Meslé (2008), who revealed a linear evolution of the life expectancy frontier using national data. Their work similarly highlights episodes of reranking, where new vanguard countries can experience faster gains before stabilizing when sticking to the frontier, with Japan being a particularly illustrative case. In our paper, we conduct the same type of analysis but at the regional level, thereby positioning our work in this lineage. We agree that the respective sentence in the ‘Results’ section requires more elaboration. However, we also feel that it is appropriately done in the discussion through the following paragraph (Line 275):

“This changing geography is crucial for understanding the potential for a longer life in Europe. Tracking life expectancy gains among the regions that constituted the vanguard in the 1990s reveals a slowing pace, and suggests possible limits to expanding human longevity. But this perspective overlooks the dynamic reranking of regions over time. When this reranking is taken into account, we observe that the pace of longevity gains has remained remarkably stable, particularly for the top-performing regions. This reinforces the thesis that human life expectancy has not yet reached its limits [8].”

7. Clarity and Context

The manuscript is generally well-written and accessible, but could benefit from clearer theoretical grounding, as noted above.

Adding more context about long-term historical trends in mortality and prior health transitions would help frame the results more robustly.

In the light of the previous comments and suggestions, we have substantially expanded the discussion section. The revised version provides now more theoretical background and a deeper discussion.

8. References

The manuscript cites relevant and up-to-date sources. However, additional references to key debates in demographic history and epidemiology (e.g., McKeown, Szreter, Omran’s transition theory) would enhance the contextual depth of the discussion.

The suggested literature has been incorporated into the discussion.

9. Expertise and Scope

I am confident assessing the demographic interpretation of mortality trends, regional analysis, and socioeconomic contextualization.

I do not have deep expertise in the specific statistical implementation of P-splines or CLM, and thus cannot fully evaluate the appropriateness of the smoothing parameter selection beyond what is reported.

We hope that our additional clarifications and the respective changes in the manuscript have improved transparency and helped to convey the rationale and robustness of our methodology.

References

Luque de Haro, V. A. (2024). Social inequalities in adult mortality across Europe (18th–21st centuries): A critical analysis of theories and evidence. *Economics & Human Biology*, 55.

<https://doi.org/10.1016/j.ehb.2024.101438>

Lutfey, K., & Freese, J. (2005). Toward some fundamentals of fundamental causality: Socioeconomic status and health in the routine clinic visit for diabetes. *American Journal of Sociology*, 110(5), 1326–1372. <https://doi.org/10.1086/428914>

Mackenbach, J. P., Valverde, J. R., Artnik, B., et al. (2018). Trends in health inequalities in 27 European countries. *Proceedings of the National Academy of Sciences*, 115(25), 6440–6445.

<https://doi.org/10.1073/pnas.1800028115>

Masselot, P., Mistry, M., Vanoli, J., et al. (2023). Excess mortality attributed to heat and cold: a health impact assessment study in 854 cities in Europe. *The Lancet Planetary Health*, 7(4), e271–e281. [https://doi.org/10.1016/S2542-5196\(23\)00023-2](https://doi.org/10.1016/S2542-5196(23)00023-2)

Thank you for suggesting these references. They are now included in the paper.

Reviewer #2 (Remarks to the Author):

Summary of Key Results

This manuscript examines changes in life expectancy from 1992 to 2019 across Western European regions. The principal strength of the study lies in the diversity of regions analyzed simultaneously, which may hold important implications for public policy if interpreted appropriately. The authors use these regions to rank gains in life expectancy and classify them as either laggards or vanguards in life expectancy improvements. The results indicate a slowdown in the pace of mortality reductions during the second half of the study period. Furthermore, the authors investigate the drivers of this deceleration in gains in e_0 , identifying the 55–74 age group as having the greatest impact on the reduction in gains.

Significance and Impact

The findings may hold considerable relevance for policymakers and researchers examining the deceleration in life expectancy improvements. The analysis is insightful, illustrating the value of considering both local and broader contextual factors when evaluating the limits of longevity.

Thank you for your very positive feedback.

The study is fully reproducible (with the exception of the dataset, which is not fully publicly available; however, this is not the responsibility of the authors). The documentation for code usage is generally thorough and accessible, except the file “O-Functions-R,” which would benefit from more detailed documentation regarding the purpose and function of each component. The code is well-structured and clear. A minor technical note is that the analysis relies on the MortalitySmooth library, which has been deprecated on CRAN. Since this requires additional steps for installation in R, it would be helpful to provide users with brief guidance on how to complete this process.

We sincerely appreciate this constructive comment and your careful attention to our code. As we are strongly committed to promoting open and reproducible science, we are especially grateful for this level of engagement.

We apologize for the lack of detailed documentation in the original version of the O-Functions.R file and for the reliance on the now-archived MortalitySmooth package, which indeed introduces unnecessary barriers for users attempting to reproduce our analysis.

In the updated OSF repository, we have addressed both concerns. Rather than providing instructions for installing archived packages, we have removed the dependency on MortalitySmooth altogether. The functions we previously relied on—mainly for GLAM arithmetic and the greedy search algorithm used for smoothing parameter selection—are now directly included in the codebase. Specifically, these are implemented in O-Functions.R and all routines have been extensively commented to clarify their purpose and logic.

We hope these improvements strengthen the transparency and accessibility of our work, and once again thank you for contributing to these enhancements through this detailed and thoughtful review.

Two methodological concerns are not addressed in the manuscript and warrant discussion. First, the issue of mixing NUTS classification levels. The NUTS classification distinguishes between NUTS-1 (major socio-economic regions), NUTS-2 (basic regions for regional policies), and NUTS-3 (small regions for specific diagnoses). The authors do not clarify why it is methodologically appropriate to compare, for example, Spain at the NUTS-3 level with Portugal at the NUTS-2 level. This inconsistency may obscure heterogeneities that the paper aims to uncover. What changes, if any, arise when the analysis is restricted to the NUTS-2 level?

Along with Reviewer 1, you raised legitimate concerns about the comparability of mortality estimates between different spatial units. We must emphasize that the selection of spatial units for our analysis was driven by data availability. Demographic statistics are typically tabulated by established administrative units, which correspond to the NUTS classification. Although the NUTS classification was created to maintain comparable regional units across the EU, substantial comparability issues could arise due to variations in population size, population density, and administrative structures of the regions at the same NUTS level. Therefore, the methodological issues you highlighted are mainly related to the limitations of the NUTS classification per se rather than the design of our study. As mentioned above, there was no other alternative for us as to use the regional mortality data available at different NUTS levels.

There was considerable variation among the European countries in terms of data availability. For some countries, the detailed regional mortality data can be easily accessed online, for the others such data might be obtained only per special request. In certain cases, the desired level of geographical details is not possible to obtain because of data protection rules. While building our dataset we aimed at the highest level of geographical detail, the longest possible time series, and the largest possible number of countries. We also kept in mind data comparability issues and the suitability of the data for visualizations, especially mortality maps. Although we believe that we ended with a decent data sample, we agree that the comparability between spatial units warrant thorough discussion. Thus, we added the following paragraph under the ‘Strengths and limitations’ section (Line 342).

“A major strength of our work lies in the unique combination of wide geographical coverage with high spatial resolution mortality series. We invested substantial effort to harmonize mortality data across regions and over time, ensuring comparability despite differences in national statistical practices (see Supplementary Table 3 for details on the implemented adjustments). This harmonization was essential to meeting our study’s central objective: assessing long-term longevity trends both by individual region and by contrasting vanguard and laggard groups. Nonetheless, perfect comparability could not be achieved. The use of the NUTS classification, though standard in European demographic research, entails considerable variation in the population size of spatial units within the same NUTS level between countries.”

Moreover, while we aimed for the most detailed geographical units and the longest possible time series, differences in data availability obliged us to use less detailed information in some countries. This averaging of outcomes across larger areas may conceal inequalities in longevity, which recent studies have shown to be important for interpreting spatial disparities [47, 48]."

Second, the use of P-splines raises concerns. Although commonly used and well-documented in demographic research, P-splines are known to perform poorly at the extremes of the age distribution, particularly at ages 0 and ω . Given the aims of the paper, overfitting at age 0 may be especially problematic, as errors at this age can propagate through the life table and significantly affect estimates of e_0 . The code suggests that some correction was applied for age 0, but this is not explained in the methods or the appendix. The authors are encouraged to elaborate on how or if this issue was addressed.

Thank you for this insightful comment, which reflects a strong understanding of the methodological foundations of our work, particularly regarding the limitations of P-splines at the extremes of the age distribution. We greatly appreciate this level of engagement.

It is indeed correct that standard P-splines can perform poorly at age 0, where mortality patterns differ markedly from the rest of the age distribution. As the reviewer observed, we addressed this issue by applying a correction for infant mortality, although we regret that this was not described in the original manuscript.

Specifically, we followed the approach proposed in Camarda (2019, Section 3.1), which modifies the basis structure over age by introducing a specialized coefficient for age 0. This coefficient is not connected to the variation of adjacent coefficients, effectively decoupling the estimation of infant mortality from that of older ages. In a one-dimensional setting, this corresponds to directly estimating the log death rate at age 0; in our two-dimensional framework, we allow this coefficient to vary smoothly over time. This strategy prevents the often-abrupt mortality level at age 0 from distorting the smooth surface over age and time, especially from age 1 onward, thereby improving both fit and interpretability.

We have now explicitly described this modelling choice in the Methodology section of the revised manuscript and added the appropriate reference to Camarda (2019), where the rationale and benefits of this approach are clearly documented.

Regarding ω , we fixed it to 95+ across all regions. This decision was made for consistency, as data for deaths or exposures were often unavailable beyond this open age group in many regions. While we acknowledge that P-spline smoothing may also be less accurate at the oldest ages, we believe that any misfit in this range has only a minimal impact on the estimates of life expectancy at birth and virtually no effect on the probability of dying between ages 55 and 74, both of which are central outcomes in our analysis.

We hope this clarification resolves the reviewer's concern, and we thank them again for highlighting this important modelling detail.

Analytical Approach

The methods employed are appropriate for addressing the research question. They are sufficiently comprehensive, and the discussion of results is both adequate and original.

We appreciate very much this positive assessment of our work.

Suggested Improvements

In the abstract, the authors state: "... is strongly associated with mortality of the population aged 55–74, which increased by 2019...". It appears the intention is to refer to the population rather than mortality increasing; the sentence is currently ambiguous.

Thank you for noting this. We replaced the phrase 'with mortality of the population aged 55–74' with 'mortality at ages 55–74'.

The rationale for using regional estimates of e_0 and comparing them should be emphasized more clearly early in the manuscript. Although the importance of this approach becomes clearer in the results and discussion sections, it would benefit from stronger justification earlier on.

As suggested, we provided the rationale for using e_0 , also compared to alternative mortality indicators, to the Introduction (Line 51). The revised text reads as follows:

"We use life expectancy at birth, a widely used and understood, easily interpretable mortality measure, as the main mortality outcome. It also has the advantage of providing more stable statistical estimates compared to partial life expectancy or life expectancy at specific ages, which is a particularly valuable feature for small spatial units. Finally, its use facilitates comparisons with other national and regional studies."

The authors explain why the analysis concludes in 2019, but do not justify why it begins in 1992. This should be addressed.

Although two distinct periods of e_0 gains are identified in Figure 1, these classifications are not applied in the subsequent analysis. Incorporating them may enhance the strength of the findings.

(Response to both comments) Our paper focuses on the analysis of long-term mortality trends for the period 1992-2019. The initial point of the analysis (1992) corresponds to the earliest year for which the data were available for all countries included in our sample. The year 2019 corresponds to the last year before the COVID-19 pandemic. Indeed, 2004-2005 reflects turning point in the trend of life expectancy gains. This point also falls very close to the mid-point of the entire period. The pandemic years were deliberately excluded from the analysis for the reasons explained in detail in our response to your last comment. As a result, we ended up with three snapshots: 1992-1993, 2004-2005, and 2018-2019. Although the respective clarification was provided in our initial submission (p.4, lines 112-116), we agree that more explanations are needed.

We would like to clarify also that the purpose of Figure 1 was not only to provide the global overview of the changes in life expectancy gains but also to justify the usage of 2004-2005 as one of the three snapshots over the observation period. These snapshots were then used consistently for Figures 3-6, as it would be cumbersome to present these figures for each biennium. By contrast, it was feasible to show the annual trends by vanguard and laggard regions in Figure 2. In Tables 1 and 2 we prefer to split the observation period 1992-2019 into three decades. This allows direct comparisons of the estimated gains with widely cited Oeppen & Vaupel (2002) paper and their '0.3 years per decade' argument, and recently, with the Olshansky et al (2024) paper defining 'radical life extension' as the average increase of 3 years per decade. Nevertheless, we agree that for sake of consistency it would be appropriate to add the results stratified by two sub-periods: before 2004-2005 and after. The respective tables replicating the results presented in Tables 1 and 2 appear now in the online supplementary material (Supplementary Tables 1 and 2).

The first sentence in the 'Results' section (Line 72) reads now as follows:

"Figure 1 offers a dual perspective on annual gains in life expectancy at birth (hereafter life expectancy or e_0) across 450 regions in Western Europe from 1992 (the earliest year for which data are available for all of the countries included) to 2019 (the last year before the COVID-19 pandemic)."

For Figure 3, the color scale should be modified to ensure readability when printed in black and white.

This could be easily implemented. However, as the journal guidelines do not state that the readability in black and white should be ensured, we used colours consistently for all figures. Should the article move to production, we are committed to adjust the figures, if it is necessary.

On page 6, line 162, the authors state: "...today's laggard regions are significantly lower than...". It should be clarified whether this refers to statistical significance.

We didn't imply statistical significance. To avoid confusion, we replaced the word 'significantly' with 'notably'.

In the Shiny app, the figures lack a color legend, which should be included

The colour legend does appear in the Shiny App. As this interactive application was primarily designed for desktop computers, the legend may not be fully visible on other devices. In some cases, the scale of the monitor has to be reduced (by pressing Ctrl-).

On page 12, line 273, the sentence "... phenomenon has only been observed in the United Kingdom..." should be nuanced to reflect that this applies only among the countries included in the study.

This sentence has been rephrased as follows (Line 299):

“Within the set of countries analysed here, this pattern has so far been documented only in the United Kingdom, with a particular concentration in northern regions.”

Questions for the authors:

Given that mortality at younger ages is relatively low in most of the regions analyzed, some may argue that other indicators, such as LE25 or LE65, may be more suitable for assessing changes in old-age mortality. Why were these not considered?

In the light of this and the previous comments we added the following paragraph to the introduction (Line 51):

“We use life expectancy at birth, a widely used and understood, easily interpretable mortality measure, as the main mortality outcome. It also has the advantage of providing more stable statistical estimates compared to partial life expectancy or life expectancy at specific ages, which is a particularly valuable feature for small spatial units. Finally, its use facilitates comparisons with other national and regional studies.”

Since the analysis aims to identify the age group contributing most to the stagnation in life expectancy gains, why was an age decomposition not performed?

We agree that an age decomposition is a valuable demographic tool. However, given the broad readership of Nature Communications, we aim to present a simpler and more accessible analysis. We believe that the approach presented in the paper is sufficient to conclude that the recent stagnation in life expectancy gains is closely linked to mortality dynamics around retirement age, which was one of our main conclusions. Therefore, while an age decomposition could provide additional detail, we consider it not essential for supporting the key messages of the study.

P-splines allow for the estimation of confidence intervals through bootstrapping. Would incorporating confidence intervals into the e_0 estimates add value to the analysis, particularly in assessing whether regional and national differences are statistically significant?

We appreciate this comment and agree that bootstrapping can be used to compute confidence intervals for e_0 and other metrics derived from P-spline models.

However, we respectfully believe that including confidence intervals for each of the 450 individual e_0 estimates would not necessarily enhance the overall value of our analysis. Our primary object of interest is the distribution of the estimated life expectancies, not the precision of individual point estimates. What matters more in our context is the aggregate pattern, not the variability around each estimate. For instance, only in Figure 2 do we present individual e_0 values for the minimum and maximum of the distribution; elsewhere, we focus directly on distributional features or on specific segments such as the upper decile. Moreover, presenting confidence intervals for all 450 regions in the main text would be cumbersome and extremely difficult, both in terms of space and interpretability. For this reason, we chose not to include confidence intervals throughout the main text.

That said, we fully agree that presenting uncertainty around selected estimates can be useful, particularly when evaluating whether differences between specific regions are statistically meaningful. To address this, we have added Supplementary Figure 5 presenting both observed and estimated life expectancy at birth for 10 French and Dutch regions, along with 95% confidence intervals computed via bootstrapping from the P-spline model. These regions present large differences in terms of population size, location and level of life expectancy. We have added a paragraph at the end of the Methodology section (Line 447) to describe this new evidence:

“To illustrate our methods, Supplementary Figure 5 presents the trend in life expectancy at birth for 10 French and Dutch regions with varying profiles (geographical location, population size, life expectancy levels) for female from 1992 to 2019. The dots represent life expectancy levels calculated using raw data, while the lines represent life expectancy levels estimated using our methods, along with 95% confidence intervals.”

In the discussion of future research, the authors suggest incorporating COVID-19 years. Why not include them in the current analysis?

As we mentioned in the original submission (p.13, line 287), focusing on the pre-pandemic period is in line with previous research addressing similar research questions and dealing with long-term mortality trends. More importantly, incorporating the pandemic years and applying our smoothing approach for the entire dataset would inevitably obscure the correct assessment of the structural trends, which was the main focus of this study. In addition, including these years would require methodological changes, since we cannot assume a smooth time trend in the presence of a sudden longevity shock.

We believe that the COVID-19 pandemic should be viewed as a short-term crisis with potentially long-term implications for mortality trends and spatial patterns. Whether this temporal shock will have a profound impact and alter regional mortality trends or these trends will return on their long-term trajectories is a subject for further investigations. For now, we would prefer to refrain from any speculations concerning this matter. To get first insights, it is essential to obtain mortality data beyond the pandemic years (at least for two-three calendar years). Evaluating regional cause-specific mortality trends and patterns should further improve our understanding of mortality trends.

In the light of this comment, we implemented several changes in the ‘Strengths and limitations’ section (Line 355):

“The extensive temporal coverage of this study is another major strength. Spanning nearly three decades prior to the COVID-19 pandemic, it offers a unique opportunity to capture the evolution of life expectancy under relatively stable conditions. By leveraging and modelling long-term mortality data, we can disentangle short-term fluctuations from structural trends, and thus provide a solid foundation for understanding the slowdown in longevity gains. Our analysis deliberately focuses on the pre-pandemic period, consistent with earlier work on long-

term mortality dynamics [1, 9]. Including the pandemic years would have inevitably interfered with our assessment of structural trends, which was the main focus of this study. Consequently, our study does not take into account the mortality burden of COVID-19, which disproportionately affected vanguard regions in 2020 (e.g., central Spain, northern Italy, and the Paris region) and laggard regions in 2021 [20]. Future research should extend the scope to incorporate both the pandemic and post-pandemic years, in order to assess whether and how this longevity crisis has reshaped trajectories of, and regional disparities in, life expectancy.”

Reviewer #3 (Remarks to the Author):

The paper presents a careful analysis of mortality and life expectancy trends across 450 subnational regions in Western Europe over a nearly 30-year period. It revealed sustained improvements in life expectancy in vanguard regions, while demonstrating different dynamics in laggard areas. This is an original, methodologically sound research, supported by beautiful visualization.

We appreciate very much your favourable assessment of our work.

I wonder whether the Introduction could be slightly extended or adjusted to better reflect the broader context of diverse trajectories in life expectancy across vanguard and laggard subpopulations.

We agree with this suggestion. The following statements have been added in the introduction (Line 60):

“These complementary concepts of vanguard and laggard populations have often been used in demographic research to describe diverse mortality trajectories at the subnational level. There has been a worldwide general trend towards widening socioeconomic health inequalities, with highly educated groups enjoying faster declines in mortality and gains in life expectancy than the groups with the lowest socioeconomic status [23]. Substantial and increasing inequalities in mortality by marital status have also been reported [24].”

In the Discussion, you refer to convergence-divergence theory and suggest that divergence may result from faster growth in vanguard regions due to earlier adoption of innovations. However, based on your results, divergence since the mid-2000s appears to be more strongly driven by a slowdown in laggard regions. It would be useful to clarify this point and consider whether the convergence-divergence framework fully captures the observed pattern.

Good point, thank you. We revised the respective paragraph that reads now as follows (Line 252):

“The driving forces behind this impressive reversal of fortunes can be better understood through the convergence–divergence framework, which explains the mechanisms leading mortality levels across populations to either converge or diverge. According to this theory, major innovations (e.g., drugs that reduce blood pressure) may initially trigger divergence, as

some countries or groups are better positioned to benefit from them. Once access broadens, convergence tends to follow [27, 28]. Importantly, convergence and divergence cycles do not necessarily alternate in a linear way; a new divergence process may unfold alongside an ongoing convergence process, complicating the interpretation of mortality trends. In our case, the convergence observed before 2005 is fully consistent with this framework. By contrast, the subsequent divergence was driven less by accelerated progress in vanguard regions than by a slowdown in laggard regions, a pattern also observed in another recent study [29]. This observation does not call into question the validity or relevance of the convergence–divergence framework, but it highlights the difficulties facing laggard regions in maintaining momentum.”

While the manuscript is descriptively rich, a more detailed discussion of the potential causal mechanisms behind the recent divergences would be beneficial. Other factors, such as differential access to healthcare, austerity measures, or economic restructuring, may be relevant beyond smoking and cardiovascular disease. These factors could be acknowledged, even if they are speculative.

Thank you for these very useful suggestions. We have added the following paragraphs to the ‘Discussion’ section (Line 306):

“Although our analyses are mainly aimed at providing novel empirical insights, and a detailed analysis of mortality determinants is beyond the scope of this study, it is important to highlight factors and potential causal mechanisms behind the observed subnational trends and differentials. Broadly speaking, it makes sense to situate the discussion of these factors within long-standing debates on the role of broad socioeconomic change versus targeted interventions in shaping mortality decline [35, 36]. The classical McKeown thesis attributes rapid mortality decline in industrialized countries to broad economic and social changes, rather than to targeted public health or medical interventions. While this controversial thesis has since been overturned on empirical grounds, it posed the fundamental question of whether population health is determined primarily by broad-based social, political and economic conditions or by more targeted policies and interventions [37].

At the subnational level, this broader debate finds a close parallel in discussions of the relative role of individual versus contextual determinants of health [38]. Accordingly, regional disparities in mortality can be interpreted in terms of compositional effects (reflecting the unequal distribution of individual risk factors across areas) or contextual effects (the influence of place-based conditions and policies). From a compositional perspective, our finding that stagnation in life expectancy has been most pronounced at ages 55–74 highlights the contribution of lifestyle-related factors such as alcohol consumption, drug use, poor diet and physical inactivity [1, 39]. At the same time, contextual factors provide important complementary explanations. Recent research has emphasized the impact of austerity measures and so-called deaths of despair (suicide, drug misuse and alcohol-specific causes) on stagnating mortality in the UK [40–42]. In our study, the regional divergence in life expectancy observed since the mid-2000s broadly coincides with the period of the 2008 financial crisis and

its aftermath, when economic growth slowed across Western Europe while becoming increasingly concentrated in major metropolitan regions [43, 44]. These urban areas, which host high-skill service industries, are now among the best-performing regions in terms of longevity. For instance, between 2005 and 2019, the western part of central London rose from 132nd to 1st place for males and from 247th to 15th for females, while Paris advanced from 40th to 32nd for males and from 45th to 4th for females.”

Although the focus on the pre-pandemic period is justified, I suggest briefly commenting on how COVID-19 might have affected the trajectories described. Even a short reflection on the potential for COVID-19 to exacerbate or alter divergence patterns would be a useful addition.

As we mentioned in the original submission (p.13, line 287), focusing on the pre-pandemic period is in line with previous research addressing similar research questions and dealing with long-term mortality trends. More importantly, incorporating the pandemic years and applying our smoothing approach for the entire dataset would inevitably obscure the correct assessment of the structural trends, which was the main focus of this study. In addition, including these years would require methodological changes, since we cannot assume a smooth time trend in the presence of a sudden longevity shock.

We believe that the COVID-19 pandemic should be viewed as a short-term crisis with potentially long-term implications for mortality trends and spatial patterns. Whether this temporal shock will have a profound impact and alter regional mortality trends or these trends will return on their long-term trajectories is a subject for further investigations. For now, we would prefer to refrain from any speculations concerning this matter. To get first insights, it is essential to obtain mortality data beyond the pandemic years (at least for two-three calendar years). Evaluating regional cause-specific mortality trends and patterns should further improve our understanding of mortality trends.

In the light of this comment, we implemented several changes in the ‘Strengths and limitations’ section (Line 355):

“The extensive temporal coverage of this study is another major strength. Spanning nearly three decades prior to the COVID-19 pandemic, it offers a unique opportunity to capture the evolution of life expectancy under relatively stable conditions. By leveraging and modelling long-term mortality data, we can disentangle short-term fluctuations from structural trends, and thus provide a solid foundation for understanding the slowdown in longevity gains. Our analysis deliberately focuses on the pre-pandemic period, consistent with earlier work on long-term mortality dynamics [1, 9]. Including the pandemic years would have inevitably interfered with our assessment of structural trends, which was the main focus of this study. Consequently, our study does not take into account the mortality burden of COVID-19, which disproportionately affected vanguard regions in 2020 (e.g., central Spain, northern Italy, and the Paris region) and laggard regions in 2021 [20]. Future research should extend the scope to incorporate both the pandemic and post-pandemic years, in order to assess whether and how this longevity crisis has reshaped trajectories of, and regional disparities in, life expectancy.”

Minor comments:

I recommend replacing “during the pre-pandemic period” with the actual years (“1992–2019”). Consider using “slowdown” or similar alternatives instead of “deceleration” (in some cases). I would also advise avoiding strong phrases like “we strongly advocate.”

Revised accordingly.

The paper opens with references to COVID-19, which is not the focus of the study. It might make more sense to start with literature on pre-pandemic stagnation in mortality improvements and only later mention the pandemic if needed.

As suggested, the literature on pre-pandemic stagnation is cited at the very beginning of the introduction.

The following might be a useful addition to the discussion of stagnating mortality improvements in other high-income countries:

Timonin, S., Leon, D. A., Banks, E., Adair, T., Canudas-Romo, V. (2024). Faltering mortality improvements at young-middle ages in high-income English-speaking countries. *International Journal of Epidemiology*, 53(3). <https://doi.org/10.1093/ije/dyae128>

Thank you for this suggestion, which supports our argument of the importance of monitoring mortality at young and adult ages. We added the following sentences to the discussion (Line 297):

“Such adverse trends at younger ages are already evident in other English-speaking countries. Indeed, it has been argued that the slowdown in life expectancy gains before the pandemic observed across these countries – with the exception of Ireland – largely stemmed from stagnating or rising mortality at young and middle adult ages [3].”

Should confidence intervals be shown, at least for regions with maximum and minimum life expectancy? Also, I couldn’t find a legend explaining the meaning of the colored lines. Using different Y-axes for males and females makes comparisons difficult and may be misleading.

We appreciate this comment and fully agree that presenting uncertainty around selected estimates can be useful, particularly when evaluating whether differences between specific regions are statistically meaningful. To address this, we have added Supplementary Figure 5 presenting both observed and estimated life expectancy at birth for 10 French and Dutch regions, along with 95% confidence intervals computed via bootstrapping from the P-spline model. These regions present large differences in terms of population size, location and level of life expectancy. We have added a paragraph at the end of the Methodology section (Line 447) to describe this new evidence:

“To illustrate our methods, Supplementary Figure 5 presents the trend in life expectancy at birth for 10 French and Dutch regions with varying profiles (geographical location, population size, life expectancy levels) for female from 1992 to 2019. The dots represent life expectancy

levels calculated using raw data, while the lines represent life expectancy levels estimated using our methods, along with 95% confidence intervals.”

Regarding the use of different Y-axes for males and females, we agree that this choice may hinder direct comparison of longevity levels between the sexes. However, the intent of the figure is not to compare males and females per se, but to highlight the differing time-trends across various parts of the life expectancy distribution. We believe that using a common Y-axis would improve comparability of levels but could obscure the heterogeneity in temporal trends, which is the key focus of this analysis.

Typically, D1 refers to the most deprived group and D10 to the least deprived (e.g., in the UK and Austria).

We understand the point made by the Reviewer, and it would be fairly easy to switch the labels for the vanguard and the laggard groups. We deliberately don't use 'least deprived' or 'most deprived' groups throughout the manuscript to avoid any parallels with the concept of socioeconomic deprivation, which was out of the scope of this study. Assigning labels to our groups was based on LE ranking. In our view, it is more intuitive to assign the first (top decile) the label D1. This is also consistent with Figure 3, where we plot the regions from 1 to 450, and from the left to the right. Finally, although many studies refer to D1 and D10 as to the most advantageous and disadvantages population groups, this is not a uniform practice (see, for example, Tetzlaff et al. 2024).

Tetzlaff F et. al. (2024). Age-specific and cause-specific mortality contributions to the socioeconomic gap in life expectancy in Germany, 2003–21: an ecological study. *The Lancet Public Health* 9 (5): e295-e305 [https://doi.org/10.1016/S2468-2667\(24\)00049-5](https://doi.org/10.1016/S2468-2667(24)00049-5) .